# GROOT: Learning to Follow Instructions by Watching Gameplay Videos

**Shaofei Cai**[1,2], **Bowei Zhang**[3], **Zihao Wang**[1,2], **Xiaojian Ma**[5], **Anji Liu**[4], **Yitao Liang**[1]*
**Team CraftJarvis**

[1]Institute for Artificial Intelligence, Peking University
[2]School of Intelligence Science and Technology, Peking University
[3]School of Electronics Engineering and Computer Science, Peking University
[4]Computer Science Department, University of California, Los Angeles
[5]Beijing Institute for General Artificial Intelligence (BIGAI)
{caishaofei,zhangbowei,zhwang}@stu.pku.edu.cn
xiaojian.ma@ucla.edu,liuanji@cs.ucla.edu,yitaol@pku.edu.cn
https://craftjarvis.github.io/GROOT

## Abstract

We study the problem of building an agent that can follow open-ended instructions in open-world environments. We propose to follow reference videos as instructions, which offer expressive goal specifications while eliminating the need for expensive text-gameplay annotations. We implement our agent GROOT in a simple yet effective encoder-decoder architecture based on causal transformers. We evaluate GROOT against open-world counterparts and human players on a proposed **Minecraft SkillForge** benchmark. The Elo ratings clearly show that GROOT is closing the human-machine gap as well as exhibiting a 70% winning rate over the best generalist agent baseline. Qualitative analysis of the induced goal space further demonstrates some interesting emergent properties, including the goal composition and complex gameplay behavior synthesis.

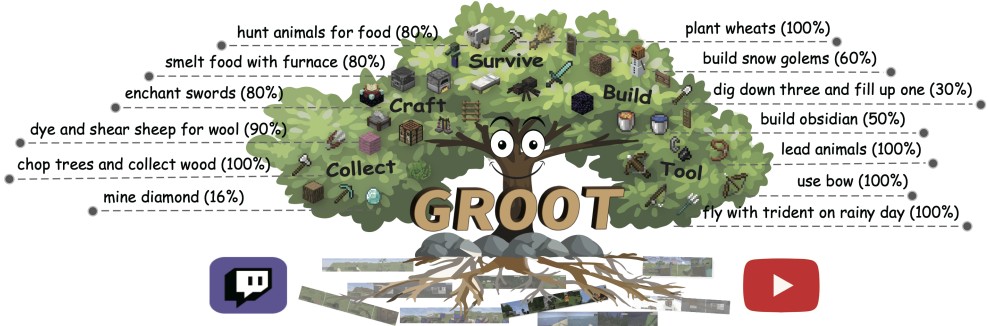

Figure 1: Through the cultivation of extensive gameplay videos, GROOT has grown a rich set of skill fruits (number denotes success rate; skills shown above do not mean to be exhaustive; kudos to our artist Haowei Lin).

## 1 Introduction

Developing human-level embodied agents that can solve endless tasks in open-world environments, such as Minecraft (Johnson et al., 2016; Fan et al., 2022), has always been a long-term goal pursued in AI. Recent works have explored using Large Language Models (LLMs) to generate high-level plans, which guide the agent to accomplish challenging long-horizon tasks (Wang et al., 2023b;a; Zhu et al., 2023). However, a major gap between these LLM-based agents and generalist agents that can

---

*Corresponding Author.

complete endless amounts of tasks is the capability of their low-level controllers, which map the plans to motor commands. Recently developed controllers are only capable of completing a predefined and narrow set of programmatic tasks (Lin et al., 2021; Baker et al., 2022; Cai et al., 2023), which hinders LLM-based planning agents from unleashing their full potential. We attribute the limitation of these low-level controllers to how the goal is specified. Specifically, existing controllers use task indicator (Yu et al., 2019), future outcome (Chen et al., 2021; Lifshitz et al., 2023), and language (Brohan et al., 2022) to represent the goal. While it is easy to learn a controller with some of these goal specifications, they may not be expressive enough for diverse tasks. Taking future outcome goals as an example, an image of a desired house clearly lacks procedural information on how the house was built. One exception is language, but learning a controller that can receive language goals is prohibitively expensive as it requires a huge number of trajectory-text pairs with text that precisely depicts the full details of the gameplay, therefore preventing them from scaling up to more open-ended tasks.

Having observed the limitations of goal specification in the prior works, this paper seeks to find a balance between the capacity of goal specification and the cost of controller learning. Concretely, we propose to specify the goal as a reference gameplay video clip. While such video instruction is indeed expressive, there are two challenges: 1) How can the controller understand the actual goal being specified as the video itself can be ambiguous, i.e., a goal space or video instruction encoder has to be learned; 2) How to ultimately map such goal to actual motor commands? To this end, we introduce a learning framework that simultaneously produces a goal space and a video instruction following controller from gameplay videos. The fundamental idea is casting the problem as future state prediction based on past observations:

- The predicting model needs to identify which goal is being pursued from the past observations, which requires a good goal space (induced by a video instruction encoder);
- Since the transition dynamics model is fixed, a policy that maps both the state and the recognized goal to action is also needed by the predicting model when rolling the future state predictions.

Effectively, this results in the goal space and control policy we need. We introduce a variational learning objective for this problem, which leads to a combination of a cloning loss and a KL regularization loss. Based on this framework, we implement GROOT, an agent with an encoder-decoder architecture to solve open-ended Minecraft tasks by following video instructions. The video encoder is a non-causal transformer that extracts the semantic information expressed in the video and maps it to the latent goal space. The controller policy is a decoder module implemented by a causal transformer, which decodes the goal information in the latent space and translates it into a sequence of actions in the given environment states in an autoregressive manner.

To comprehensively evaluate an agent's mastery of skills, we designed a benchmark called **Minecraft SkillForge**. The benchmark covers six common Minecraft task groups: `collect`, `build`, `survive`, `explore`, `tool`, and `craft`, testing the agent's abilities in resource collection, structure building, environmental understanding, and tool usage, in a total of 30 tasks. We calculate Elo ratings among GROOT, several counterparts, and human players based on human evaluations. Our experiments showed that GROOT is closing the human-machine gap and outperforms the best baseline by 150 points (or 70% winning rate) in an Elo tournament system. Our qualitative analysis of the induced goal space further demonstrates some interesting emergent properties, including the goal composition and complex gameplay behavior synthesis.

To sum up, our main contributions are as follows:

- Start by maximizing the log-likelihood of future states given past ones, we have discovered the learning objectives that lead to a good goal space and ultimately the instruction-following controller from gameplay videos. It provides theoretical guidance for the agent architecture design and model optimization.
- Based on our proposed learning framework, we implemented a simple yet efficient encoder-decoder agent based on causal transformers. The encoder is responsible for understanding the goal information in the video instruction while the decoder as the policy emits motor commands.
- On our newly introduced benchmark, Minecraft SkillForge, GROOT is closing the human-machine gap and surpassing the state-of-the-art baselines by a large margin in the overall Elo rating comparison. GROOT also exhibits several interesting emergent properties, including goal composition and complex gameplay behavior synthesis.

## 2 PRELIMINARIES AND PROBLEM FORMULATION

Reinforcement Learning (RL) concerns the problem in which an agent interacts with an environment at discrete time steps, aiming to maximize its expected cumulative reward (Mnih et al., 2015; Schulman et al., 2017; Espeholt et al., 2018). Specifically, the environment is defined as a Markov Decision Process (MDP) $\langle \mathcal{S}, \mathcal{A}, \mathcal{R}, \mathcal{P}, d_0 \rangle$, where $\mathcal{S}$ is the state space, $\mathcal{A}$ is the action space, $\mathcal{R} : \mathcal{S} \times \mathcal{A} \to \mathbb{R}$ is the reward function, $\mathcal{P} : \mathcal{S} \times \mathcal{A} \to \mathcal{S}$ is the transition dynamics, and $d_0$ is the initial state distribution. Our goal is to learn a policy $\pi(a|s)$ that maximizes the expected cumulative reward $\mathbb{E}[\sum_{t=0}^{\infty} \gamma^t r_t]$, where $\gamma \in (0, 1]$ is a discount factor.

In goal-conditioned RL (GCRL) tasks, we are additionally provided with a goal $g \in \mathcal{G}$ (Andrychowicz et al., 2017; Ding et al., 2019; Liu et al., 2022; Cai et al., 2023; Jing et al., 2021; 2020; Yang et al., 2019). And the task becomes learning a goal-conditioned policy $\pi(a|s, g)$ that maximizes the expected return $\mathbb{E}[\sum_{t=0}^{\infty} \gamma^t r_t^g]$, where $r_t^g$ is the goal-specific reward achieved at time step $t$. Apart from being a new type of RL task, GCRL has been widely studied as a pre-training stage toward conquering more challenging environments/tasks (Aytar et al., 2018b; Baker et al., 2022; Zhang et al., 2022). Specifically, suppose we are provided with a good goal-condition policy, the goal can be viewed as a meta-action that drives the agent to accomplish various sub-tasks, which significantly simplifies tasks that require an extended horizon to accomplish. Further, when equipped with goal planners, we can achieve zero- or few-shot learning on compositional tasks that are beyond the reach of RL algorithms (Huang et al., 2022; Wang et al., 2023b;a; Zhu et al., 2023; Gong et al., 2023).

At the heart of leveraging such benefits, a key requirement is to have a properly-defined goal space that (i) has a wide coverage of common tasks/behaviors, and (ii) succinctly describes the task without including unnecessary information about the state. Many prior works establish goal spaces using guidance from other modalities such as language (Hong et al., 2020; Stone et al., 2023; Cai et al., 2023) or code (Wang et al., 2023a; Huang et al., 2023). While effective, the requirement on large-scale trajectory data paired with this auxiliary information could be hard to fulfill in practice. Instead, this paper studies the problem of simultaneously learning a rich and coherent goal space and the corresponding goal-conditioned policy, given a pre-trained inverse dynamic model and raw gameplay videos, i.e., sequences of states $\{s_{1:T}^{(i)}\}_i$ collected using unknown policies.

## 3 GOAL SPACE DISCOVERY VIA FUTURE STATE PREDICTION

This section explains our learning framework: discovering a "good" goal space as well as a video instruction following the controller through the task of predicting future states given previous ones. We start with an illustrative example in Minecraft (Johnson et al., 2016). Imagine that an agent is standing inside a grassland holding an axe that can be used to chop the tree in front of them. Suppose in the gameplay video, players either go straight to chop the tree or bypass it to explore the territory. In order to predict future frames, it is sufficient to know (i) which goal (chop tree or bypass tree) is being pursued by the agent, and (ii) what will happen if the agent chooses a particular option (i.e., transition dynamics). Apart from the latter information that is irrelevant to the past observations, we only need to capture the goal information, i.e., whether the agent decides to chop the tree or bypass the tree. Therefore, the task of establishing a comprehensive yet succinct goal space can be interpreted as predicting future states while conditioning on the transition dynamics of the environment.

Formally, our learning objective is to maximize the log-likelihood of future states given past ones: $\log p_\theta(s_{t+1:T}|s_{1:t})$. Define $g$ as a latent variable conditioned on past states (think of it as the potential goals the agent is pursuing given past states), the evidence lower-bound of the objective given variational posterior $q_\phi(g|s_{1:T})$ is the following (see Appendix A for derivations):

$$\log p_\theta(s_{t+1:T}|s_{1:t}) = \log \sum_g p_\theta(s_{t+1:T}, g|s_{1:t})$$

$$\geq \mathbb{E}_{g \sim q_\phi(\cdot|s_{1:T})} \left[ \log p_\theta(s_{t+1:T}|s_{1:t}, g) \right] - D_{\mathrm{KL}} \left( q_\phi(g|s_{1:T}) \parallel p_\theta(g|s_{1:t}) \right),$$

where $D_{\mathrm{KL}}(\cdot\|\cdot)$ denotes the KL-divergence. Next, we break down the first term (i.e., $p_\theta(s_{t+1:T}|s_{1:t}, g)$) into components contributed by the (unknown) goal-conditioned policy $\pi_\theta(a|s, g)$

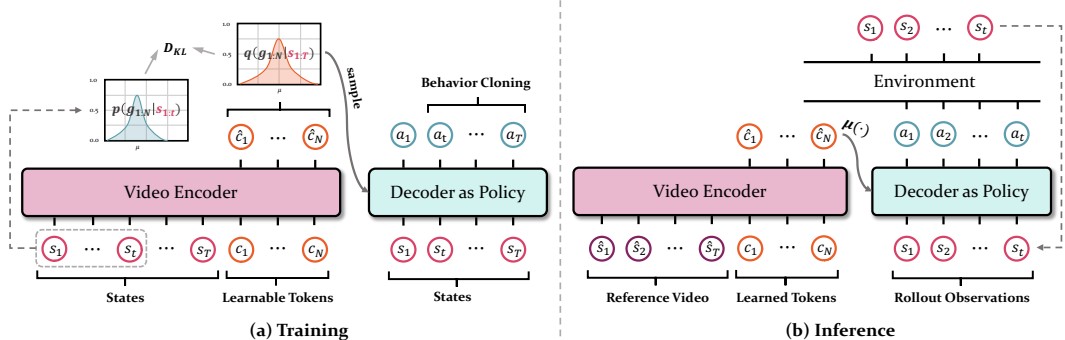

Figure 2: **GROOT agent architecture. Left:** In the training stage, a video encoder (non-causal transformer) learns to extract the semantic meaning and transfer the video (state sequence) into the goal embedding space. A goal-conditioned policy (causal transformer) is learned to predict actions following the given instructions. We learn the agent using behavior cloning under a KL constraint. **Right:** During inference, a reference video is passed into the encoder to generate the goal embeddings that drive the policy to interact with the environment.

and the transition dynamics $p_\theta(s_{t+1}|s_{1:t}, a_t)$ :

$$\log p_\theta(s_{t+1:T}|s_{1:t}, g) = \sum_{\tau=t}^{T} \log \sum_{a_\tau} \pi_\theta(a_\tau|s_{1:\tau}, g) \cdot p_\theta(s_{\tau+1}|s_{1:\tau}, a_\tau)$$

$$\geq \sum_{\tau=t}^{T} \mathbb{E}_{a_\tau \sim p_\theta(a_\tau|s_{1:\tau+1})} \big[ \log \pi_\theta(a_\tau|s_{1:\tau}, g) + C \big],$$

where the constant $C$ contains terms that depend solely on the environment dynamics and are irrelevant to what we want to learn (i.e., the goal space and the goal-conditioned policy). Bring it back to the original objective, we have

$$\log p(s_{t+1:T}|s_{1:t}) \geq \underbrace{\sum_{\tau=t}^{T-1} \mathbb{E}_{g \sim q_\phi(\cdot|s_{1:T}), a_\tau \sim p_\theta(\cdot|s_{1:\tau+1})} [\log \pi_\theta(a_\tau|s_{1:\tau}, g)]}_{\text{behaviour cloning}} - \underbrace{D_{\mathrm{KL}} (q_\phi(g|s_{1:T}) \parallel p_\theta(g|s_{1:t}))}_{\text{goal space constraint (KL regularization)}},$$

where $q_\phi(\cdot|s_{1:T})$ is implemented as a video encoder that maps the whole state sequence into the latent goal space. $p_\theta(\cdot|s_{1:\tau+1})$ is the inverse dynamic model (IDM) that predicts actions required to achieve a desired change in the states, which is usually a pre-trained model, details are in Appendix C. Thus, the objective can be explained as jointly learning a video encoder and a goal-controller policy through behavior cloning under succinct goal space constraints.

## 4 GROOT ARCHITECTURE DESIGN AND TRAINING STRATEGY

This section illustrates how to create an agent (we call it GROOT) that can understand the semantic meaning of a reference video and interact with the environment based on the aforementioned learning framework. According to the discussion in Section 3, the learnable parts of GROOT include the video encoder and the goal-conditioned policy. Recently, Transformer (Vaswani et al., 2017) has demonstrated effectiveness in solving sequential decision-making problems (Parisotto et al., 2019; Chen et al., 2021; Brohan et al., 2022). Motivated by this, we implement GROOT with transformer-based encoder-decoder architecture, as shown in Figure 2.

### 4.1 VIDEO ENCODER

The video encoder includes a Convolutional Neural Network (CNN) to extract spatial information from image states $s_{1:T}$ and a non-causal transformer to capture temporal information from videos. Specifically, we use a CNN backbone to extract visual embeddings $\{x_{1:T}\}$ for all frames. Additionally, motivated by Devlin et al. (2019); Dosovitskiy et al. (2020), we construct a set of learnable embeddings (or summary tokens), represented as $\{c_{1:N}\}$, to capture the semantic information present in the video.

The visual embeddings and summary tokens are passed to a non-causal transformer, resulting in the output corresponding to the summary tokens as $\{\hat{c}_{1:N}\}$

$$x_{1:T} \leftarrow \texttt{Backbone}(s_{1:T}),$$
$$\hat{c}_{1:N} \leftarrow \texttt{Transformer}([x_{1:T}, c_{1:N}]). \tag{1}$$

Similar to VAE (Kingma & Welling, 2013), we assume that the latent goal space follows a Gaussian distribution, hence we use two fully connected layers, $\mu(\cdot)$ and $\sigma(\cdot)$, to generate the mean and standard deviation of the distribution, respectively. During training, we use the reparameterization trick to sample a set of embeddings $\{g_{1:N}\}$ from the distribution, where $g_t \sim \mathcal{N}(\mu(\hat{c}_t), \sigma(\hat{c}_t))$. During inference, we use the mean of the distribution as the goal embeddings, i.e. $g_t \leftarrow \mu(\hat{c}_t)$.

## 4.2 DECODER AS POLICY

To introduce our policy module, we start with VPT (Baker et al., 2022), a Minecraft foundation model trained with standard behavioral cloning. It is built on Transformer-XL (Dai et al., 2019) that can leverage long-horizon historical states and predict the next action seeing the current observation. However, the vanilla VPT architecture does not support instruction input. To condition the policy on goal embeddings, we draw the inspiration from Flamingo (Alayrac et al., 2022), that is, to insert *gated cross-attention dense* layers into every Transformer-XL block. The keys and values in these layers are obtained from goal embeddings, while the queries are derived from the environment states

$$\hat{x}_{1:t}^{(l)} \leftarrow \texttt{GatedXATTN}(kv = g_{1:N}, q = x_{1:t}^{(l-1)}; \theta_l),$$
$$x_{1:t}^{(l)} \leftarrow \texttt{TransformerXL}(qkv = \hat{x}_{1:t}^{(l)}; \theta_l), \tag{2}$$
$$\hat{a}_t \leftarrow \texttt{FeedForward}(x_t^{(M)}),$$

where the policy reuses the visual embeddings extracted by the video encoder, i.e., $x_{1:t}^{(0)} = x_{1:t}$, the policy consists of $M$ transformer blocks, $\theta_l$ is the parameter of $l$-th block, $\hat{a}_t$ is the predicted action. Since our goal space contains information about how to complete a task that is richer than previous language-conditioned policy (Cai et al., 2023; Lifshitz et al., 2023), the cross-attention mechanism is necessary. It allows the GROOT to query the task progress from instruction information based on past states, and then perform corresponding behaviors to complete the remaining progress.

## 4.3 TRAINING AND INFERENCE

The training dataset can be a mixture of Minecraft gameplay videos and offline trajectories. For those videos without actions, an inverse dynamic model (Baker et al., 2022) can be used to generate approximate actions. Limited by the computation resources, we truncated all the trajectories into segments with a fixed length of $T$ without using any prior. We denote the final dataset as $\mathcal{D} = \{(x_{1:T}, a_{1:T})\}_M$, where $M$ is the number of trajectories. We train GROOT in a fully self-supervised manner while the training process can be viewed as self-imitating, that is, training GROOT jointly using behavioral cloning and KL divergence loss

$$\mathcal{L}(\theta, \phi) = \mathbb{E}_{\substack{(s_{1:T}, a_{1:T}) \sim \mathcal{D} \\ g \sim q_\phi(\cdot|s_{1:T})}} \left[ \sum_{\tau=t}^{T-1} -\log \pi_\theta(a_\tau|s_{1:\tau}, g) + \lambda_{KL} D_{KL} \left( q_\phi(g|s_{1:T}) \parallel p_\theta(g|s_{1:t}) \right) \right], \tag{3}$$

where $\lambda_{KL}$ is the tradeoff coefficient, $q_\phi$ is a posterior visual encoder, $p_\theta$ is a prior video encoder with the same architecture. More details are in the Appendix D.

## 5 RESULT

### 5.1 PERFORMANCE ON MASTERING MINECRAFT SKILLS

**Minecraft SkillForge Benchmark.** In order to comprehensively evaluate the mastery of tasks by agents in Minecraft, we created a diverse benchmark called **Minecraft SkillForge**. It covers 30 tasks from 6 major categories of representative skills in Minecraft, including `collect`, `explore`, `craft`, `tool`, `survive`, and `build`. For example, the task "dig three down and fill one up" in the `build` category asks the agent to first dig three blocks of dirt, then use the dirt to fill the space above; The task of "building a snow golem" (🧙) requires the agent to sequentially stack 2 snow

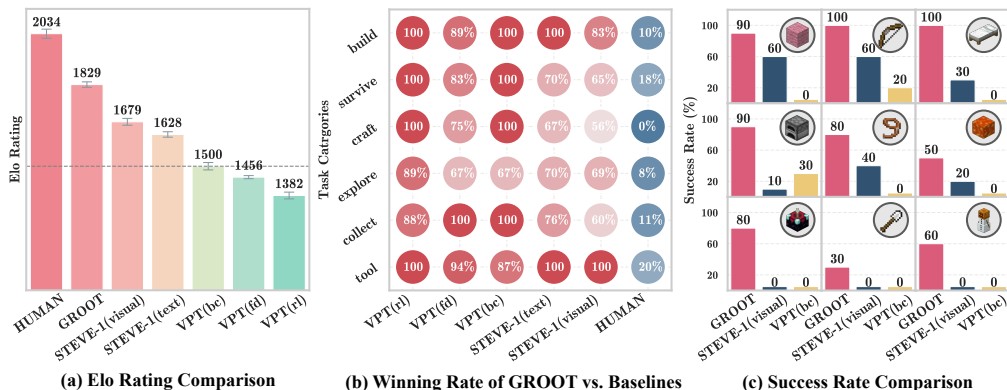

(a) Elo Rating Comparison     (b) Winning Rate of GROOT vs. Baselines     (c) Success Rate Comparison

Figure 3: **Results on Minecraft SkillForge benchmark.** **Left:** Tournament evaluation of GROOT assessed by human players. GROOT performs better than state-of-the-art Minecraft agent STEVE-1. A 150-score gap corresponds to a 70% probability of winning. **Middle:** Winning rate of GROOT v.s. other agents on specific task categories. Colors from red to blue denote a decrease in the winning rate. Apart from the human player, GROOT surpasses all other baselines. **Right:** Success rate on 9 representative tasks. GROOT champions process-oriented tasks, such as dig three down and fill one up (🗝) and build snow golems (⛄).

blocks (⬜) and 1 carved pumpkin (🎃). We put the details of this benchmark in the Appendix H. Apart from some relatively simple or common tasks such as "collect wood" and "hunt animals", other tasks require the agent to have the ability to perform multiple steps in succession.

We compare GROOT with the following baselines: (a) VPT (Baker et al., 2022), a foundation model pre-trained on large-scale YouTube data, with three variants: VPT (fd), VPT(bc), and VPT(rl), indicating vanilla foundation model, behavior cloning finetuned model, and RL finetuned model; (b) STEVE-1 (Lifshitz et al., 2023), an instruction-following agent finetuned from VPT, with two variants: STEVE-1 (visual) and STEVE-1 (text) that receives visual and test instructions. More details are in Appendix F.1. *It is worth noting that GROOT was trained from scratch.*

**Human Evaluation with Elo Rating.** We evaluated the relative strength of agents by running an internal tournament and reporting their Elo ratings, as in Mnih et al. (2015). Before the tournament, each agent is required to generate 10 videos of length 600 on each task. Note that, all the reference videos used by GROOT are generated from another biome to ensure generalization. Additionally, we also invited 3 experienced players to do these tasks following the same settings. After the video collection, we asked 10 players to judge the quality of each pair of sampled videos from different agents. Considering the diversity of tasks, we designed specific evaluation criteria for every task to measure the quality of rollout trajectories. After 1500 comparisons, the Elo rating converged as in Figure 3 (left). Although there is a large performance gap compared with human players, GROOT has significantly surpassed the current state-of-the-art STEVE-1 series and condition-free VPT series on the overall tasks. Additional details are in Appendix G.

In Figure 3 (middle), we compare GROOT with other baselines in winning rate on six task groups. We found that except for the performance on craft tasks, where STEVE-1 (visual) outperforms our model, GROOT achieves state-of-the-art results. In particular, GROOT greatly outperforms other baselines by a large margin on build and tool. For build, the goal space needs to contain more detailed procedural information, which is the disadvantage of methods that use future outcomes as the goal. Moreover, such tasks are distributed sparsely in the dataset, or even absent in the dataset, which requires the agent to have strong generalization ability. As for craft group, GROOT is not superior enough, especially on the "crafting table" task. We attribute this to the wide task distribution in the dataset. Thus the future outcomes can prompt STEVE-1 to achieve a high success rate.

**Programmatic Evaluation.** To quantitatively compare the performance of the agents, we selected 9 representative tasks out of 30 and reported the success rate of GROOT, STEVE-1 (visual), and VPT (bc) on these tasks in Figure 3 (right). We found that, based on the success rate on tasks such as dye and shear sheep(🟥), enchat sword(⚔), smelt food(🍖), use bow (🏹), sleep (🛏), and lead animals (🪢), GROOT has already reached a level comparable to that of human players (100%). However, the success rates for build snow golems (⛄) and build obsidian (🟫) tasks are only 60% and 50%. By observing the generated videos, we

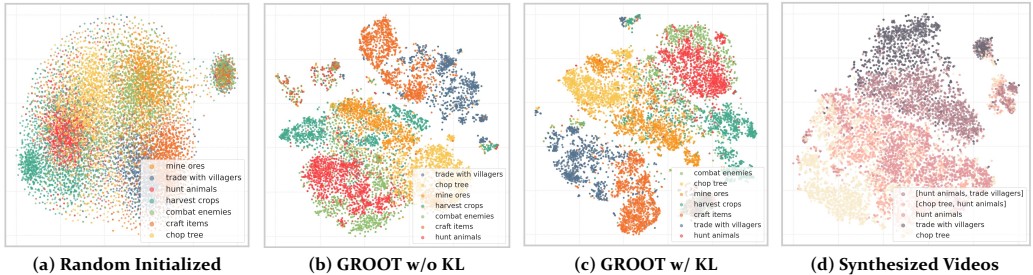

**(a) Random Initialized**     **(b) GROOT w/o KL**     **(c) GROOT w/ KL**     **(d) Synthesized Videos**

Figure 4: **t-SNE visualization of the goal space.** Each color corresponds to a specific video category. **(Left):** Space of randomly initialized video encoder. All the videos are entangled together. **Middle:** Space of GROOT trained with self-supervised learning w/ and w/o KL regularization, respectively. The videos are clustered based on their semantics. Visualization shows the subtle differences between the two. **Right:** Synthesized videos using concatenation manner. The concatenated videos lay on the position between the source videos.

found that GROOT cannot precisely identify the items in Hotbar (such as buckets, lava buckets, snow blocks, and pumpkin heads), resulting in a low probability of switching to the correct item. STEVE-1 also has the same problem. This may be due to the current training paradigm lacking strong supervisory signals at the image level. Future work may introduce auxiliary tasks such as vision-question answering (VQA) to help alleviate this phenomenon. Details are in Appendix F.3.

## 5.2 PROPERTIES OF LEARNED GOAL SPACE

This section studies the properties of learned goal space. We used the t-SNE algorithm (van der Maaten & Hinton, 2008) to visualize the clustering effect of reference videos encoded in goal space, as in Figure 4. We select 7 kinds of videos, including `craft items`, `combat enemies`, `harvest crops`, `hunt animals`, `chop trees`, `trade with villagers`, and `mine ores`. These videos are sampled from the contractor data (Baker et al., 2022) according to the meta information (details are in Appendix F.2). Each category contains 1k video segments. As a control group, in Figure 4 (left), we showed the initial goal space of the video encoder (with a pre-trained EfficientNet-B0 (Tan & Le, 2019) as the backbone) before training. We found that the points are entangled together. After being trained on offline trajectories, as in Figure 4 (middle), it well understands reference videos and clusters them according to their semantics. This proves that it is efficient to learn behavior-relevant task representations using our self-supervised training strategy. The clustering effect is slightly better with KL regularization, though the difference is not very significant. Inevitably, there are still some videos from different categories entangled together. We attribute this to the possibility of overlap in the performed behaviors of these videos. For example, `chop trees` and `harvest crops` both rely on a sequential of "attack" actions.

**Condition on Concatenated Videos.** We also study the possibility of conditioning the policy on concatenated videos. First, we collect 3 kinds of source videos, including `chop trees`, `hunt animals`, and `trade with villagers`. We randomly sampled two videos from sources of `chop trees` and `hunt animals`, downsampled and concatenated them into a synthetic video, denoted as [`chop trees, hunt animals`]. By the same token, we can obtain [`hunt animals, trade with villagers`]. We visualize these videos together with the source videos in Figure 4 (right). We found that the source videos lie far away from each other while the concatenated videos are distributed between their source videos. Based on this intriguing phenomenon, we infer that concatenated videos may prompt GROOT to solve both tasks simultaneously. To verify this, we evaluate GROOT on three kinds of reference videos, i.e., `chop trees`, `hunt animals`, and [`chop trees, hunt animals`]. We launched GROOT in the forest and in the animal plains, respectively. The collected wood and killed mobs are reported in Figure 5. We found that although the concatenated video may not be as effective as raw video in driving an agent to complete a single task (60% of the performance of raw video), it does possess the ability to drive the agent to perform multiple tasks. This is an important ability. As discussed in Wang et al. (2023b), sometimes the high-level planner will propose multiple candidate goals, it will be efficient if the low-level controller can automatically determine which to accomplish based on the current observation.

**Ablation on KL Divergence Loss.** To investigate the role of KL loss in training, we evaluated GROOT (w/ KL) and its variant (w/o KL) on three tasks: `collect seagrass` (🌿), `collect`

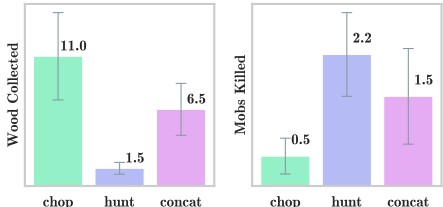

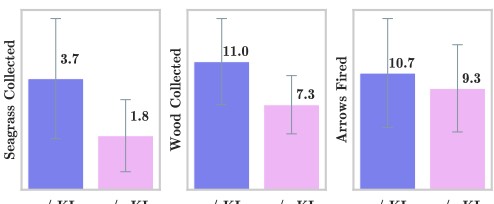

Figure 5: **Comparison on using raw and concatenated reference videos as conditions. Left:** Collected wood in the forest biome. **Right:** Killed mobs in the plains biome. "concat" denotes the reference video is [chop trees, hunt animals]. Statistics are measured over 10 episodes.

Figure 6: **Ablation study on KL loss.** After being jointly trained with KL loss, GROOT can collect $2\times$ more seagrass (🌿) underwater and $1.5\times$ wood (🪵) in the forest while the difference is not as impressive on the use bow (🏹) task. Statistics are measured over 10 episodes.

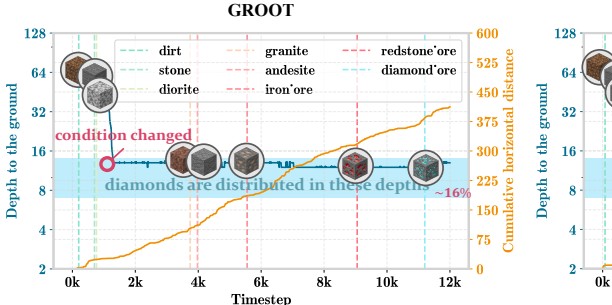

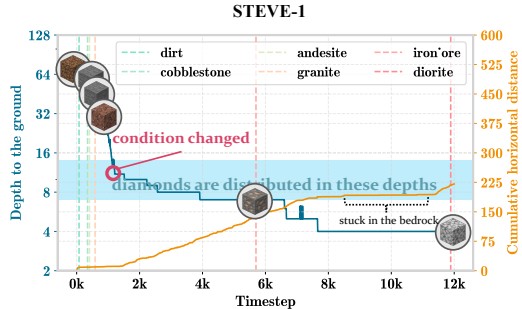

Figure 7: **Results on solving challenging `obtain diamond` task.** The vertical dashed lines represent the time when a certain item is first obtained. **Left:** GROOT first dug down to the depth of 12 and then mined horizontally to obtain diamonds with an average success rate of $16\%$. **Right:** STEVE-1 quickly dug down to the specific depth but struggled to maintain its height.

wood (🪵), and use bow (🏹). As shown in Figure 6, we found that introducing the constraint of KL loss improved agent performance by $2\times$ and $1.5\times$ in the first two tasks, whereas there was no significant effect in the use bow task. This may be because the first two tasks require the agent to generalize the corresponding skills to different terrains (e.g. locating trees in the environment for collecting wood and sinking to specific locations for collecting seagrass). Therefore, it puts higher demands on the agent's ability to generalize in the goal space, and this is exactly the role played by the KL loss. The use bow task is relatively simple in comparison because it only requires charging and shooting the arrow, without considering environmental factors.

## 5.3 COMBINING SKILLS FOR LONG-HORIZON TASKS

In this section, we explore whether GROOT can combine skills to solve long-horizon tasks, which is key to its integration with a high-level planner. Taking the task of mining diamonds as an example, prior knowledge is that diamond ores are generally distributed between the 7th and 14th floors underground, and the probability of appearing in other depths is almost zero. Therefore, the agent needs to first dig down to the specified depth (12) and then maintain horizontal mining. To achieve this, we designed two reference videos, each 128 frames long. One describes the policy of starting from the surface and digging down, and the other demonstrates the behaviors of horizontal mining. We show an example in Figure 7 (left). In the beginning, GROOT quickly digs down to the specified depth and then switches to horizontal mining mode. It maintains the same height for a long time and found diamonds at 11k steps. In addition, we compared STEVE-1 (visual) under the same setting in Figure 7 (right). After switching to the horizontal mining prompt, STEVE-1 maintains its height for a short time before it stuck in the bedrock layer (unbreakable in survival mode), greatly reducing the probability of finding diamonds. This indicates that our goal space is expressive enough to instruct the way of mining, and the policy can follow the instructions persistently and reliably. In contrast, STEVE-1, which relies on future outcomes as a condition, was unable to maintain its depth, despite attempts at various visual prompts. We conducted 25 experiments each on GROOT and STEVE-1, with success rates of $16\%$ and $0\%$ for finding diamonds. Additional details are in the Appendix F.4.

## 6 RELATED WORKS

**Pre-train Policy on Offline Data.** Pre-training neural networks on web-scale data has been demonstrated as an effective training paradigm in Nature Language Processing (Brown et al., 2020) and Computer Vision (Kirillov et al., 2023). Inspired by this, researchers tried to transfer the success to the field of decision-making from pre-training visual representations and directly distilling the policy from offline data. As the former, Aytar et al. (2018a); Bruce et al. (2023) leveraged temporal information present in videos as the supervision signal to learn visual representations. The representations are then used to generate intrinsic rewards for boosting downstream policy learning, which still requires expensive online interactions with the environment. Schmidhuber (2019); Chen et al. (2021) leveraged scalable offline trajectories to train optimal policy by conditioning it on cumulated rewards. Laskin et al. (2022) proposed to learn an in-context policy improvement operator that can distill an RL algorithm in high data efficiency. Reed et al. (2022) learned a multi-task agent Gato by doing behavior cloning on a large-scale expert dataset. By serializing task data into a flat of sequence, they use the powerful transformer architecture to model the behavior distribution. However, these methods either require elaborated reward functions or explicit task definitions. This makes it hard to be applied to open worlds, where tasks are infinite while rewards are lacking. Another interesting direction is to use pre-trained language models for reasoning and vision language models for discrimination, to guide the policy in life-long learning in the environment (Di Palo et al., 2023).

**Condition Policy on Goal Space.** Researchers have explored many goal modalities, such as language (Khandelwal et al., 2021), image (Du et al., 2021), and future video (Xie et al., 2023), to build a controllable policy. Brohan et al. (2022) collected a large-scale dataset of trajectory-text pairs and trained a transformer policy to follow language instructions. Despite the language being a natural instruction interface, the cost of collecting paired training data is expensive. As a solution, Majumdar et al. (2022) sorted to use hindsight relabeling to first train a policy conditioned on the target image, then aligned text to latent image space, which greatly improves training efficiency. Lifshitz et al. (2023) moved a big step on this paradigm by replacing the target image with a 16-frame future video and reformulating the modality alignment problem into training a prior of latent goal given text.

**Build Agents in Minecraft.** As a challenging open-world environment, Minecraft is attracting an increasing number of researchers to develop AI agents on it, which can be divided into plan-oriented (Wang et al., 2023b;a) and control-oriented methods (Baker et al., 2022; Cai et al., 2023; Lifshitz et al., 2023) based on their emphasis. Plan-oriented agents aim to reason with Minecraft knowledge and decompose the long-horizon task into sub-tasks followed by calling a low-level controller. Control-oriented works follow the given instructions and directly interact with the environments using low-level actions (mouse and keyboard). Baker et al. (2022) pre-trained the first foundation model VPT in Minecraft using internet-scale videos. Although it achieves the first obtaining diamond milestone by fine-tuning with RL, it does not support instruction input. Lifshitz et al. (2023) created the first agent that can solve open-ended tasks by bridging VPT and MineCLIP (Fan et al., 2022). However, its goal space is not expressive enough and prevents it from solving multi-step tasks.

## 7 LIMITATIONS AND CONCLUSION

Although GROOT has demonstrated powerful capabilities in expressing open-ended tasks in the form of video instructions, training such a goal space remains highly challenging. We found that GROOT is quite sensitive to the selection of reference videos, which we attribute to the fact that the goal space trained from an unsupervised perspective may not be fully aligned with the human intention for understanding the semantics of the reference video. Therefore, it would be a promising research direction in the future to use SFT (supervised fine-tuning, Sanh et al. (2021)) and RLHF (Ziegler et al., 2019) to align the pre-trained goal space with human preference.

We propose a paradigm for learning to follow instructions by watching gameplay videos. We prove that video instruction is a good form of goal space that not only expresses open-ended tasks but can be trained through self-imitation (once the IDM is available to label pseudo actions for raw gameplay videos). Based on this, we built an encoder-decoder transformer architecture agent named GROOT in Minecraft. Without collecting any text-video data, GROOT demonstrated extraordinary instruction-following ability and crowned the Minecraft SkillForge benchmark. Additionally, we also demonstrate its potential as a planner downstream controller in the challenging `obtain diamond` task. We believe that this training paradigm can be generalized in other complex open-world environments.

ACKNOWLEDGEMENTS

This work is funded in part by the National Key R&D Program of China #2022ZD0160301, a grant from CCF-Tencent Rhino-Bird Open Research Fund. We thank Haowei Lin for contributing the thumbnail figure of GROOT.

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

# APPENDIX

## A  DERIVATION

In this section, we detail how we derive the final objective. Recall that the goal is to maximize the log-likelihood of future states given past ones: $\log p(s_{t+1:T}|s_{1:t})$. Using Bayes' theorem and the Jensen's inequality, we have:

$$\log\ p(s_{t+1:T}|s_{1:t}) = \log \sum_z p(s_{t+1:T}, z|s_{1:t}), \tag{4}$$

$$= \log \sum_z \frac{p(s_{t+1:T}, z|s_{0:t})\, q(z|s_{1:T})}{q(z|s_{1:T})}, \tag{5}$$

$$\geq \mathbb{E}_{z\sim q(z|s_{1:T})}\big[\log\ p(s_{t+1:T}, z|s_{1:t}) - \log\ q(z|s_{1:T})\big], \tag{6}$$

$$= \mathbb{E}_{z\sim q(z|s_{1:T})}\big[\log\ p(s_{t+1:T}|s_{1:t}, z) + \log\ p(z|s_{1:t}) - \log q(z|s_{1:T})\big], \tag{7}$$

$$= \mathbb{E}_{z\sim q(z|s_{1:T})}\big[\log\ p(s_{t+1:T}|s_{1:t}, z)\big] + \mathbb{E}_{z\sim q(z|s_{1:T})}\big[\log \frac{p(z|s_{1:t})}{q(z|s_{1:T})}\big], \tag{8}$$

$$= \mathbb{E}_{z\sim q(z|s_{1:T})}\big[\log\ p(s_{t+1:T}|s_{1:t}, z)\big] - D_{\text{KL}}\big(q(z|s_{1:T}) \,\|\, p(z|s_{1:t})\big). \tag{9}$$

We break down $p(s_{t+1:T}|s_{1:t}, z)$ into components: goal-conditioned policy $\pi(a_\tau|s_{1:\tau+1})$ and the transition dynamics $p(s_{t+1}|s_{1:t}, a_t)$, we have

$$p(s_{t+1:T}|s_{1:t}, z) = \prod_{\tau=t}^{T-1} \big(\sum_{a_\tau} \pi(a_\tau|s_{0:\tau}, z) \cdot p(s_{\tau+1}|s_{1:\tau}, a_\tau)\big). \tag{10}$$

Furthermore, using Jensen's inequality, $\log p(s_{t+1:T}|s_{0:t}, z)$ can be written as

$$\log\ p(s_{t+1:T}|s_{1:t}, z) = \sum_{\tau=t}^{T-1} \log \sum_{a_\tau} \pi(a_\tau|s_{1:\tau}, z) \cdot p(s_{\tau+1}|s_{1:\tau}, a_\tau), \tag{11}$$

$$= \sum_{\tau=t}^{T-1} \log \sum_{a_\tau} \pi(a_\tau|s_{1:\tau}, z) \cdot \frac{p(a_\tau|s_{1:\tau}, s_{\tau+1}) \cdot p(s_{\tau+1}|s_{1:\tau})}{p(a_\tau|s_{1:\tau})}, \tag{12}$$

$$\geq \sum_{\tau=t}^{T-1} \mathbb{E}_{a_\tau\sim p(a_\tau|s_{1:\tau}, s_{\tau+1})}\big[\log \pi(a_\tau|s_{1:\tau}, z) + C\big], \tag{13}$$

where the constant $C = \log p(s_{\tau+1}|s_{1:\tau}) - \log p(a_\tau|s_{1:\tau})$ describes the dataset distribution and is irrelevant to what we want to learn (i.e., the goal space and the goal-conditioned policy), we have:

$$\mathbb{E}_{z\sim q(z|s_{1:T})}\big[\log p(s_{t+1:T}|s_{1:t}, z)\big] \geq \mathbb{E}_{z\sim q(z|s_{1:T})}\big[\sum_{\tau=t}^{T-1} \mathbb{E}_{a_\tau\sim p(a_\tau|s_{1:\tau}, s_{\tau+1})}\big[\log \pi(a_\tau|s_{1:\tau}, z)\big]\big], \tag{14}$$

$$= \sum_{\tau=t}^{T-1} \mathbb{E}_{z\sim q(z|s_{1:T}), a_\tau\sim p(a_\tau|s_{1:\tau}, s_{\tau+1})}\big[\log \pi(a_\tau|s_{1:\tau}, z)\big]. \tag{15}$$

Thus, we derived the evidence lower-bound of $\log p(s_{t+1:T}|s_{1:t})$ as follows

$$\log p(s_{t+1:T}|s_{1:t}) \geq \sum_{\tau=t}^{T-1} \mathbb{E}_{z\sim q(z|s_{1:T}), a_\tau\sim p(a_\tau|s_{1:\tau+1})}\big[\log \pi(a_\tau|s_{1:\tau}, z)\big] - D_{\text{KL}}\big(q(z|s_{1:T}) \,\|\, p(z|s_{1:t})\big). \tag{16}$$

## B  MINECRAFT ENVIRONMENT

Minecraft is an extremely popular sandbox game that allows players to freely create and explore their world. This game has infinite freedom, allowing players to change the world and ecosystems through building, mining, planting, combating, and other methods (shown in Figure 8). It is precisely because

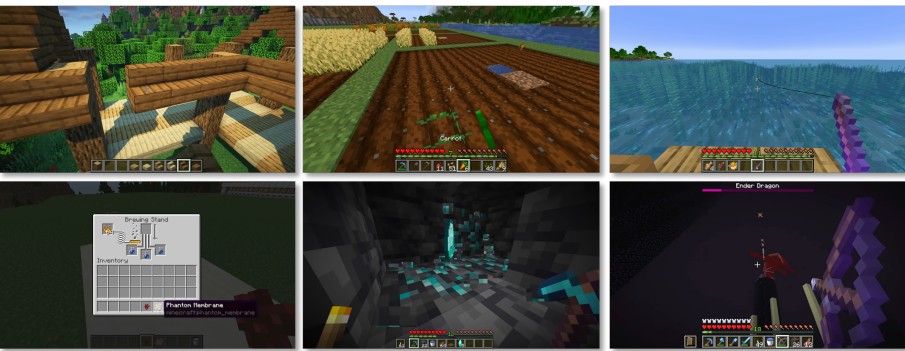

Figure 8: Examples of Minecraft environment. Tasks from top to bottom, from left to right are building houses, planting wheat, fishing, brewing a potion, mining diamond ores, and combating the ender dragon, respectively.

of this freedom that Minecraft becomes an excellent AI testing benchmark (Johnson et al., 2016; Baker et al., 2022; Fan et al., 2022; Cai et al., 2023; Lifshitz et al., 2023; Wang et al., 2023b;a). In this game, AI agents need to face situations that are highly similar to the real world, making judgments and decisions to deal with various environments and problems. Therefore, Minecraft is a very suitable environment to be used as an AI testing benchmark. By using Minecraft, AI researchers can more conveniently simulate various complex and diverse environments and tasks, thereby improving the practical value and application of AI technology.

We use the combination of 1.16.5 version MineRL (Guss et al., 2019) and MCP-Reborn[1] as our testing platform, which is consistent with the environment used by VPT (Baker et al., 2022) and STEVE-1 (Lifshitz et al., 2023). Mainly because this platform preserves observation and action space that is consistent with human players to the fullest extent. On the one hand, this design brings about high challenges, as agents can only interact with the environment using low-level mouse and keyboard actions, and can only observe visual information like human players without any in-game privileged information. Therefore, the AI algorithms developed on this platform can have higher generalization ability. On the other hand, this also presents opportunities for us to conduct large-scale pre-training on internet-scale gameplay videos.

### B.1 OBSERVATION SPACE

The visual elements included in our observation space are completely consistent with those seen by human players, including the Hotbar, health indicators, player hands, and equipped items. The player's perspective is in the first person with a field of view of 70 degrees. The simulator first generates an RGB image with dimensions of $640 \times 360$ during the rendering process. Before inputting to the agent, we resize the image to $224 \times 224$ to enable the agent to clearly see item icons in the inventory and important details in the environment. When the agent opens the GUI, the simulator also renders the mouse cursor normally. The RGB image is the only observation that the agent can obtain from the environment during inference. It is worth noting that to help the agent see more clearly in extremely dark environments, we have added a night vision effect for the agent, which increases the brightness of the environment during nighttime.

### B.2 ACTION SPACE

Our action space is almost identical to that of humans, except for actions that involve inputting strings. It consists of two parts: the mouse and the keyboard. The mouse movement is responsible for changing the player's camera perspective and moving the cursor when the GUI is opened. The left and right buttons are responsible for attacking and using items. The keyboard is mainly responsible for controlling the agent's movement. We list the meaning of each action in the Table 1. To avoid predicting null action, we used the same joint hierarchical action space as Baker et al. (2022), which consists of button space and camera space. Button space encodes all combinations of keyboard operations and a flag indicating whether the mouse is used, resulting in a total of 8461

---

[1]https://github.com/Hexeption/MCP-Reborn

Table 1: Action space descriptions from Minecraft wiki (https://minecraft.fandom.com/wiki/Controls).

| Index | Action | Human Action | Description |
|---|---|---|---|
| 1 | Forward | key W | Move forward. |
| 2 | Back | key S | Move backward. |
| 3 | Left | key A | Strafe left. |
| 4 | Right | key D | Strafe right. |
| 5 | Jump | key Space | Jump. When swimming, keeps the player afloat. |
| 6 | Sneak | key left Shift | Slowly move in the current direction of movement. When used in conjunction with the attack function in the GUI, it can swap items between inventory and Hotbar. When used with the craft function, it crafts the maximum possible number of items instead of just one. |
| 7 | Sprint | key left Ctrl | Move quickly in the direction of current motion. |
| 8 | Attack | left Button | Destroy blocks (hold down); Attack entity (click once). |
| 9 | Use | right Button | Put down the item being held or interact with the block that the player is currently looking at. Within the GUI, pick up a stack of items or place a single item from the stack that is being held by the mouse. |
| 10 | hotbar.[1-9] | keys 1 - 9 | Selects the appropriate hotbar item. When in the inventory GUI, swap the contents of the inventory slot under the mouse pointer and the corresponding hotbar slot. |
| 11 | Yaw | move Mouse X | Turning; aiming; camera movement.Ranging from -180 to +180. |
| 12 | Pitch | move Mouse Y | Turning; aiming; camera movement.Ranging from -180 to +180. |

candidate actions. The camera space discretizes the range of one mouse movement into 121 actions. Therefore, the action head of the agent is a multi-classification network with 8461 dimensions and a multi-classification network with 121 dimensions.

## C  INVERSE DYNAMIC MODEL

According to the theory in Section 3, we know that our training paradigm relies on the inverse dynamic model (IDM) which generates pseudo action labels for raw gameplay videos to calculate the behavior cloning loss. Therefore, in this section, we introduced the background knowledge of IDM.

IDM is a non-causal model that aims to uncover the underlying action that caused changes in the current step by observing historical and future states, and it can be formally represented as $p(a_t|o_t, o_{t+1})$. Compared with traditional policies learned via behavior cloning, IDM is more accurate in predicting actions because it can observe the changes between past and future frames. OpenAI (Baker et al., 2022) developed the first inverse dynamic model in the Minecraft domain. By extending the length of the observable sequence to 128 and modeling $p(a_t|o_{t-64:t+64})$ with a non-causal transformer, the IDM achieved the accuracy of action prediction to over $95\%$ with only 2k hours of game trajectories. This makes it possible for our training paradigm to utilize the large-scale Minecraft data available on the Internet. Moreover, Zhang et al. (2022) has also trained an accurate IDM model with a small amount of data in a real autonomous driving environment, which further provides a basic guarantee for our training method to generalize to other complex environments.

## D  IMPLEMENTATION DETAILS

### D.1  MODEL ARCHITECTURE

The video encoder consists of a convolutional neural network backbone and a non-causal transformer. Inspired by Brohan et al. (2022), we adopted the EfficientNet (Tan & Le, 2019) as the backbone. Specifically, we use its variant EfficientNet-B0 for efficiency, which takes in images of size $224 \times 224$

and extracts a feature vector of shape $7 \times 7 \times 1280$, where $7 \times 7$ denotes the spatial dimensions. In order to adaptively enhance the important visual information, we use a shallow transformer to pool the feature map along spatial channels. To fuse global visual features, we construct another learnable embedding `[sp]`, concatenate it with the 49 features in space, and obtain a token sequence of length 50. After being processed by the transformer, the output for the `[sp]` token corresponds to the pooled visual feature, whose dimension is $d_{hid} = 1024$. To capture the temporal features of the video, we remove the code related to the casual mask in the minGPT[2] and obtain a non-causal transformer. The policy decoder consists of 4 identical blocks, where each block contains a Flamingo *gated-attention dense layer* (Alayrac et al., 2022) and a Transformer-XL block(Dai et al., 2019). The Transformer-XL block maintains a recurrence memory of past 128 key-value pairs to memory long-horizon history states. We directly use the Transformer-XL implementation in Baker et al. (2022) with a simple modification, i.e., before passing states into the policy decoder, we add the previous action to the state embedding at each timestep. Notably, We find this modification **very useful** especially when we need to train the policy from scratch. As it not only accelerates the training process but makes the predicted action more **consistent** and **smooth**. Additional hyperparameters can be found in Table 2.

Table 2: Hyperparameters for training GROOT.

| Hyperparameter | Value |
| --- | --- |
| Optimizer | AdamW |
| Weight Decay | 0.001 |
| Learning Rate | 0.0000181 |
| Warmup Steps | 2000 |
| Number of Workers | 4 |
| Parallel Strategy | ddp |
| Type of GPUs | NVIDIA RTX 4090Ti, A40 |
| Parallel GPUs | 8 |
| Accumulate Gradient Batches | 8 |
| Batch Size/GPU (Total) | 2 (128) |
| Training Precision | bf16 |
| Input Image Size | $224 \times 224$ |
| CNN Backbone | EfficientNet-B0 |
| Encoder Transformer | minGPT (w/o causal mask) |
| Decoder Transformer | TransformerXL |
| Number of Encoder Blocks | 8 |
| Number of Decoder Blocks | 4 |
| Hidden Dimension | 1024 |
| Number of Condition Slots | 1 |
| Trajectory Chunk size | 128 |
| Attention Memory Size | 256 |
| Weight of KL Loss | 0.01 |

### D.2 INFERENCE

To generate reference videos, we invited three human players to play each task according to the task description. Each person was asked to produce two videos, so we could prepare six videos for each task in total. Then, we selected the most relevant video to the task description from the six videos and cropped the first 128 frames into a new video, which was used to instruct GROOT to complete this task. In addition, we selected a 16-frame segment that best expressed the task information as the visual prompt for STEVE-1 (visual) from these six videos. This ensures fairness in comparison.

During inference, we found that in some tasks, such as `build obsidian` (⬢), GROOT's behavior mixed with the intention of traveling around. We believe this is a bias introduced during training. We draw the inspiration from STEVE-1 (Lifshitz et al., 2023) and subtract this bias in the action logits space before sampling the action. Specifically, we infer two models at the same time, where one model's condition is a specific task video and the other model's condition is a 128-frame

---
[2]https://github.com/karpathy/minGPT

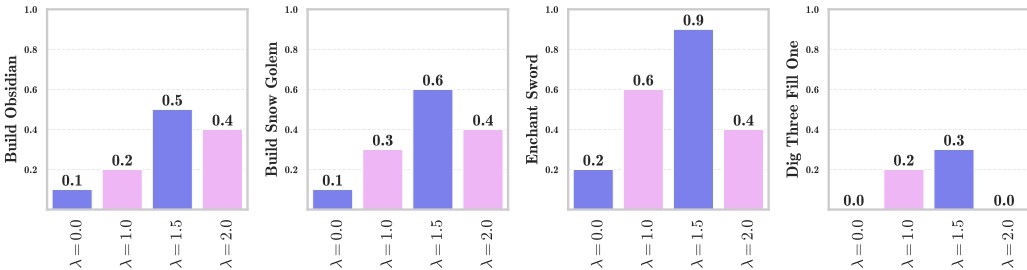

Figure 9: **Ablation on the condition scale $\lambda$.**

video of traveling freely in the environment. The input observations for the two models are exactly the same. At each time step, we use the action logits of the previous model to subtract a certain proportion of the action logits predicted by the latter model before using the Gumbel-Softmax trick to sample the action. The logits calculation equation is directly borrowed from Lifshitz et al. (2023)

$$\text{logits}_t = (1 + \lambda) f_\theta(o_{1:t}, g_{\text{goal}}) - \lambda f_\theta(o_{1:t}, g_{\text{bias}}) \tag{17}$$

where $f_\theta(o_{1:t}, g_{\text{goal}})$ and $f_\theta(o_{1:t}, g_{\text{bias}})$ are two kinds of action logits generated by feeding forward two reference videos goal and bias to GROOT, $\lambda$ is a trade-off parameter. As illustrated in Figure 9, we find that this trick can improve the success rate of tasks such as build obsidian (⬛), build snow golem (🗿), enchant sword (⚔), and dig three down and fill one up (🔨) with the $\lambda = 1.5$. Interestingly, we observe that the effective $\lambda$ scale (approximately 1.5) in our model is much smaller than the scale (approximately 6.5) used in STEVE-1. We speculate that this may be because STEVE-1 fine-tunes the foundation VPT to gain steerability, but VPT does not receive goal conditions for demonstrations during behavior cloning. This may cause VPT to learn overly smooth behavior distributions, requiring the use of larger lambda scales to activate goal-specific behaviors. Although this technique is effective at inference time, it still requires hyperparameter tuning in practice. In the future, it will be meaningful to directly remove biased behaviors from the training process.

## D.3 ABLATION ON NUMBER OF CONDITION SLOTS

In this section, we explore the impact of the number of condition slots (number of learnable tokens) on the final performance. We compared the performance of the model on 6 programmatic tasks with $N = 1$ and $N = 3$ condition slots and computed quantitative metrics for each task. As shown in Table 3, we find that increasing the number of condition slots leads to a significant decrease in the model's performance on most tasks, except for the "explore run" task. We speculate that having more condition slots may result in a higher number of dimensions in the goal space, which in turn reduces the generalization ability of the learned goal space. Therefore, we suggest that when applying GROOT to other environments, the hyperparameters should be carefully chosen based on the characteristics of the environment or using other parameter selection methods.

Table 3: **Ablation on the number of condition slots.**

| Task Name (Metric) | explore run ↑ (distance) | build pillar ↓ (height of pillar) | collect grass ↓ (num of grass) | collect seagrass ↓ (num of seagrass) | collect dirt ↓ (num of dirt) | mine stone ↓ (num of stones) |
|---|---|---|---|---|---|---|
| $N = 1$ | 54.0 | 37.6 | 23.8 | 3.3 | 6.2 | 12.2 |
| $N = 3$ | 59.0 | 13.3 | 5.6 | 0.9 | 5.4 | 11.2 |

## E DATASET DETAILS

### E.1 CONTRACTOR DATA

The contractor data is a Minecraft offline trajectory dataset provided by Baker et al. (2022) [3], which is annotated by professional human players and used for training the inverse dynamic model. In this

---

[3] https://github.com/openai/Video-Pre-Training

dataset, human players play the game while the system records the image sequence $\{s_{1:T}\}_M$, action sequence $\{a_{1:T}\}_M$, and metadata $\{e_{1:T}\}_M$ generated by the players. Excluding frames containing empty actions, the dataset contains 1.6 billion frames with a duration of approximately 2000 hours. The metadata records the events triggered by the agent in the game at each time step, including three types: `craft item`, `pickup`, and `mine block`, which represent the agent's activities of crafting items using the GUI, picking up dropped items and destroying blocks at the current time step, respectively. *In the process of training GROOT, we use all trajectories provided by the contractor data, but without including any metadata. We only use the metadata to retrieve relevant trajectory segments during the visualization of the goal space.*

## F  Experimental Setup Details

### F.1  Baseline Details

VPT is the first foundation model in the Minecraft domain developed by Baker et al. (2022). Its architecture consists of ImpalaCNN and TransformerXL. Using behavior cloning algorithms to pre-train on large-scale YouTube demonstrations, they obtained the first checkpoint of VPT(fd) which can freely explore the environment. To further enhance the agent's abilities in early-game environments, they constructed an "earlygame" dataset and fine-tuned the pre-trained foundation model on that dataset, resulting in the VPT(bc) checkpoint. This model significantly improved performance on basic tasks such as "crafting table" and "collecting wood". Based on VPT(bc), they used online reinforcement learning with a carefully designed reward shaping to obtain the checkpoint VPT(rl) capable of obtaining diamonds entirely from scratch. *It is noteworthy that the models' architectures of all three checkpoints are consistent and do not support instruction input.* That's why their rankings on the Minecraft SkillForge benchmark are low. We also observed that the performance of VPT(bc) surpasses that of VPT(rl) due to the "earlygame" dataset's exploratory nature, making it perform better on `explore` tasks. VPT(rl) is tailored specifically for diamond mining tasks and has thus lost the capability of most tasks outside diamond mining path. No matter where you place it, the first thing VPT(rl) does is to look for trees and prepare to mine diamonds.

STEVE-1 is a Minecraft agent that can follow open-ended text and visual instructions built on MineCLIP (Fan et al., 2022) and VPT. It can perform a wide range of short-horizon tasks that can be expressed by a 16-frame future video clip. The training of STEVE-1 can be described in two steps. The first step is to train a future-video conditioned policy with packed hindsight relabeling trick. With the frozen MineCLIP visual encoder to embed the visual instruction, they finetune the VPT(bc) on the contractor data to obtain STEVE-1(visual). The second step is to learn a model that translates textual instruction into visual instruction. By training a conditional variational autoencoder (CVAE) on the collected video-text pairs, they created a variant STEVE-1(text) that understands text instructions. This baseline performs well on many simple tasks in the Minecraft SkillForge benchmark, such as "explore run," "collect grass," and "collect wood." However, it struggles with multi-step and less common tasks, like "build snow golems" and "dig three down and fill one up."

*Please note that all baselines, including GROOT, were not fine-tuned for tasks in Minecraft SkillForge.*

### F.2  t-SNE Visualization Details

This section details how the videos are sampled to do visualization. The selected videos are categorized into seven groups: `craft items`, `combat enemies`, `harvest crops`, `hunt animals`, `chop trees`, `trade with villagers`, and `mine ores`. Generally, each group contains two types of videos, each with 1000 data points sampled. The sampling method retrieves the time when a certain event occurs in the metadata and goes back 128 frames from that time to obtain a video segment that is 128 frames long. We illustrate video configurations in Table 4. For example, in the `combat enemies` task, taking "combat zombies" as an example, we retrieve all the moments when the event "pickup:rotten_flesh" occurs, because after killing zombies, they will drop rotten flesh, which can then be picked up by players. Through sampling observations, we found that this method can sample videos that are consistent with the descriptions.

Table 4: Sample videos from the contractor data (Baker et al., 2022) for the goal space visualization.

| Group | Video Description | Event in Metadata |
|---|---|---|
| craft items | craft wodden_pickaxe with crafting_table | craft_item:wooden_pickaxe |
| craft items | craft iron_pickaxe with crafting_table | craft_item:iron_pickaxe |
| combat enemies | combat zombies | pickup:rotten_flesh |
| combat enemies | combat spiders | pickup:spider_eye |
| harvest crops | harvest wheat | mine_block:wheat |
| harvest crops | harvest melon | mine_block:melon |
| hunt animals | hunt sheep | pickup:mutton |
| hunt animals | hunt cow | pickup:beef |
| chop trees | chop oak trees | mine_block:oak_log |
| chop trees | chop birch trees | mine_block:birch_log |
| trade with villagers | trade with villagers for emerald | craft_item:emerald |
| trade with villagers | trade with villagers for enchanted_book | craft_item:enchanted_book |
| mine ores | mine coal ores with pickaxe | mine_block:coal_ore |
| mine ores | mine iron ores with pickaxe | mine_block:iron_ore |

### F.3 PROGRAMMATIC EVALUATION DETAILS

In this section, we elaborated on how each episode is regarded as successful. For the `dye and shear sheep` (🟥) task, dyeing the sheep and shearing its wool must be successfully performed to be considered a success. For the `use bow` (🏹) task, firing the arrow after charging it to the maximum degree is required to be successful. For the `sleep` (🛏) task, placing the bed and spending the night on it are required to be successful. For the `smelt` (⬛) task, placing the furnace and dragging coal and mutton into the designated slots are required to be successful. For the `lead` (🪢) task, successfully tethering at least one animal is considered a success. For the `build obsidian` (🟪) task, pouring a water bucket and a lava bucket to fuse them is required to be successful. For the `enchant` (📖) task, placing the enchantment table, putting a diamond sword and lapis lazuli into the slots, and clicking the enchanting option are required to be successful. For the `dig down three fill one up` (🔑) task, the agent must first vertically break three dirt blocks below and then use one dirt block to seal the area above. For the `build snow golems` (⛄) task, placing 2 snow blocks and 1 carved pumpkin head in order and triggering the creation of a snow golem are required to be successful.

### F.4 COMBINING SKILLS EXPERIMENTAL DETAILS

First, we introduce the experimental environment selected for our study. The agent is summoned on the plains biome, holding a diamond pickaxe, and granted the night vision status to enable the agent to see the various ores underground. At the beginning of each episode, we set the agent's condition to `dig down`. When the agent descends to a depth below 12 layers, the condition automatically switches to `horizontal mining`. Each round of episodes lasts for 12,000 frames, which is equivalent to 10 minutes in the real world. For GROOT, both the reference videos of `dig down` and `horizontal mining` were recorded by a human player. For STEVE-1, we invited the same player to carefully record the prompt videos. It is worth noting that while we could easily prompt it to dig down, it was difficult to keep it in the horizontal mining condition. This made STEVE-1 prone to falling into the bedrock layer and getting stuck. Finally, we did not observe STEVE-1 finding any diamonds in the 25 experiments, which can be attributed to the inability of its goal space to encode details such as horizontal mining.

## G RATING SYSTEM

### G.1 ELO RATING

The ELO rating system is widely adopted for evaluating the skill levels of multiple players in two-player games, such as Chess and Go (Silver et al., 2016). In this section, we elaborate on how we

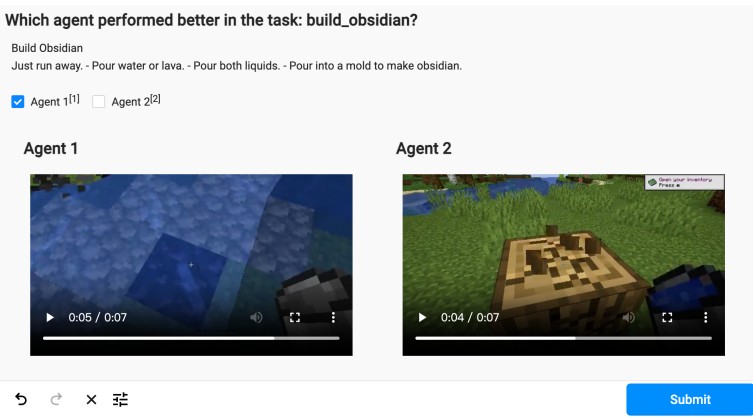

Figure 10: Example of the annotating system for human evaluation.

introduce human evaluation and use the ELO Rating system to measure the relative performance of agents on the Minecraft SkillForge benchmark.

In the ELO rating system, each agent's skill level is represented by a numerical rating. We repeatedly let agents play against each other in pairs. Specifically, in each game, we sample a task and two agents, denoted as Agent A and Agent B. Then, we randomly sample a trajectory for each agent corresponding to the designated task. The two trajectories are assigned to a human annotator, who selects the most task-relevant one. We implement the annotating system with Label Studio (Tkachenko et al., 2020-2022), as shown in Figure 10. We consider the agent that produced this trajectory to be the winner, let's assume it is Agent A. After each round, we update the scores of Agent A and Agent B as follows

$$
\begin{aligned}
R_A &\leftarrow R_A + K \cdot \frac{1}{1 + 10^{(R_A - R_B)/400}}, \\
R_B &\leftarrow R_B - K \cdot \frac{1}{1 + 10^{(R_A - R_B)/400}},
\end{aligned}
\tag{18}
$$

where K is the update factor and we set it to 8. After calculating the score of the agent, we use VPT (bc) as 1500 points and shift the scores of other agents accordingly. Based on the ELO ratings, we can easily measure the relative winning rate for each paired agent. The win rate of Agent A over Agent B can be represented as $\frac{1}{1+10^{(R_B-R_A)/400}}$. For example, the win rate ratio between two agents with a score difference of 100 scores is $64\% : 36\%$. A score difference of 200 scores implicit $76\% : 24\%$.

## G.2 TrueSkill Rating

We also report the comparison results using TrueSkill [4] rating system, which is used by gamers to evaluate their skill level. It was developed by Microsoft Research and is currently used on Xbox LIVE for matchmaking and ranking services. Different from ELO, it can also track the uncertainty of the rankings. This system utilizes the Bayesian inference algorithm to quantify a player's true skill points. In TrueSkill, rating is modeled as a Gaussian distribution which starts from $\mathcal{N}(25, \frac{25}{3}^2)$, where $\mu$ is an average skill of player, and $\sigma$ is a confidence of the guessed rating. A real skill of player is between $\mu \pm 2\sigma$ with $95\%$ confidence. After conducting 1500 updates, the TrueSkill scores converged as in Table 5. We found that the ranking order of the baseline methods is consistent with that obtained using ELO rating: HUMAN $\succ$ GROOT $\succ$ STEVE-1(visual) $\succ$ STEVE-1(text) $\succ$ VPT(bc) $\succ$ VPT(fd) $\succ$ VPT(rl).

Table 5: TrueSkill rating comparison on the Minecraft SkillForge benchmark.

| Baseline | HUMAN | GROOT | STEVE-1(visual) | STEVE-1(text) | VPT(bc) | VPT(fd) | VPT(rl) |
|---|---|---|---|---|---|---|---|
| $\mu \pm \sigma$ | $34.2 \pm 1.0$ | $29.1 \pm 0.9$ | $25.8 \pm 0.8$ | $24.6 \pm 0.8$ | $22.2 \pm 0.8$ | $20.7 \pm 0.8$ | $19.2 \pm 0.9$ |

---

[4]https://trueskill.org/

### G.3 Human Participation

We recruited 15 students with varying degrees of Minecraft game experience, ranging from a few hours to several years, from the Minecraft project group to conduct the evaluation. They are all familiar with the basic operations of Minecraft. Each employee was asked to label 100 matches for ELO Rating or TrueSkill Rating, for a total of 1500 matches. For each employee who is required to collect or assess gameplay videos, we ask them to first read the description of each task in the Minecraft SkillForge Benchmark completely, as well as the evaluation criteria for task completion quality, see Appendix H. Taking the task of building a snow golem as an example, the evaluation criteria are as follows: *Build a snow golem. ≻ Place both kinds of blocks. ≻ Place at least one kind of block. ≻ Place no block.* This enables employees to quantify video quality and ensures that all employees evaluate task completion consistently. *All these employees were explicitly informed that the collected data would be used for AI research.*

## H MINECRAFT SKILLFORGE BENCHMARK

In this section, we detail the benchmark titled "Minecraft SkillForge" which meticulously incorporates a wide spectrum of tasks prevalent within Minecraft. Our aim is to ensure that every task provides a meaningful evaluation of a specific skill that an AI agent might possess. We categorize these tasks into six groups: `collect`, `explore`, `craft`, `tool`, `survive`, and `build`. In the following subsections, we will provide a detailed introduction to each of them. The "Description" field provides a brief description of the task, the "Precondition" field outlines the initial settings of the testing environment for the task, the "SkillAssessed" field indicates which aspect(s) of the agent's ability are being assessed by the task, and the "Evaluation" field describes the quality evaluation metrics for task completion (based on which human players judge the quality of two rollout videos).

### H.1 COLLECT

The tasks categorized under the `collect` section of our benchmark are specifically designed to evaluate an AI agent's capability in resource acquisition proficiency and spatial awareness. This means the agent should not only be adept at identifying and gathering specific resources but also possess the acumen to navigate through varied environments while being aware of its surroundings and the available tools at its disposal.

```
Task: collect dirt
Description: Collect dirt from the surface.
Precondition: Spawn the player in the plains biome.
SkillAssessed: Basic terrain understanding and the ability to differentiate
    between surface-level blocks.
Evaluation: Run away. < Look down. < Dig down. < Break the dirt on the surface.

Task: collect grass
Description: Remove weeds on the surface.
Precondition: Spawn the player in the plains biome.
SkillAssessed: Surface navigation and comprehension of vegetation blocks.
Evaluation: Run away. < Break the grass block. < Break a large field of grass
    blocks.

Task: collect wood
Description: Cut down trees to collect wood.
Precondition: Spawn the player in the forest biome with an iron_axe in its hand.
SkillAssessed: Recognition of tree structures, efficient utilization of tools, and
     block harvesting capability.
Evaluation: Run away. < Approach trees. < Chop the tree and collect logs.

Task: collect seagrass
Description: Dive into the water and collect seagrass.
Precondition: Spawn the player near the sea.
SkillAssessed: Water navigation, diving mechanics understanding, and underwater
    block interaction.
Evaluation: Walk on the land. < Swim on the water < Dive into the water. < Break
    seagrass blocks.

Task: collect wool
```

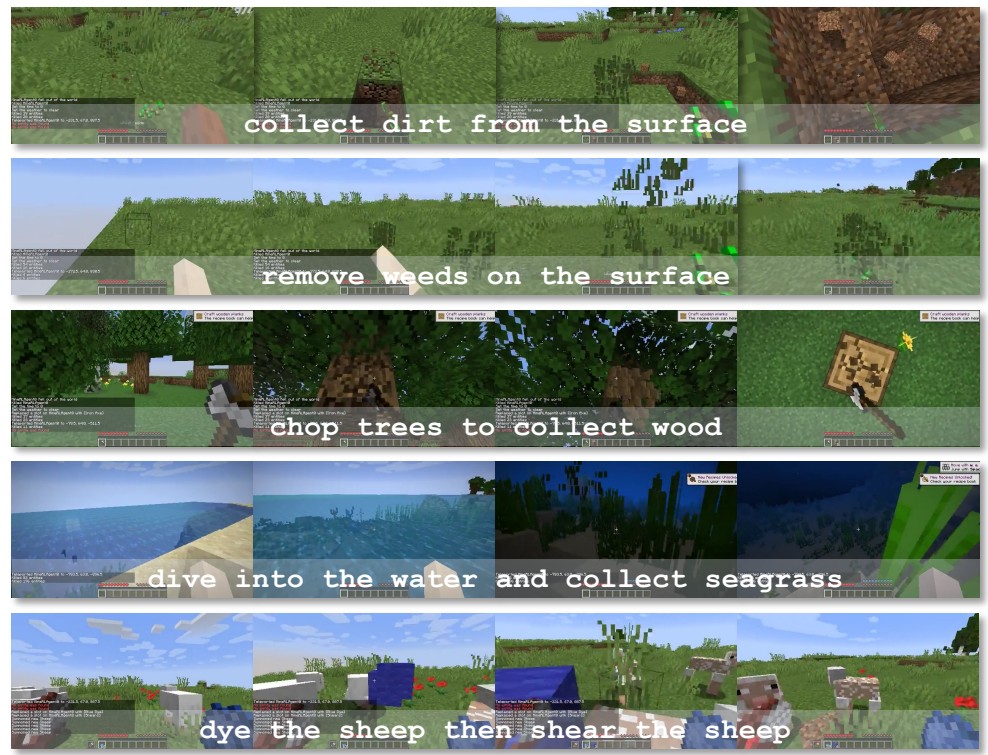

Figure 11: Examples of tasks in `collect` category.

```
Description: Dye and shear the sheep for wool.
Precondition: Spawn the player in the plains biome with a shear (mainhand) and a
    stack of blue_dye (offhand), 5 sheep near the player.
SkillAssessed: Interaction with entities, tool and item application, and
    sequential action execution.
Evaluation: Ignore the sheep. < Dye the sheep. < Shear the sheep. < First dye then
    shear the sheep.
```

Listing 1: The environment configuration and evaluation metric for `collect` series tasks.

## H.2 EXPLORE

The tasks encompassed within the `explore` category of our benchmark are intricately devised to evaluate an AI agent's navigation proficiency, understanding of diverse environments, and intrinsic motivation for exploration. Through these tasks, we gauge an agent's ability to actively traverse, understand, and interact with varied elements of the Minecraft world, and its propensity to unravel mysteries and challenges posed by the environment.

```
Task: run and explore
Description: Run and explore.
Precondition: Spawn the player in the plains biome.
SkillAssessed: Stamina utilization and distance-based exploration.
Evaluation: Exploring as far as possible.

Task: climb the mountain
Description: Climb the mountain.
Precondition: Spawn the player in the stone shore biome and near the mountain.
SkillAssessed: Vertical navigation, terrain adaptation, and goal-oriented movement.

Evaluation: Run away and ignore the mountain. < Approach the mountain. < Climbing
    the mountain. < Climb to the top of the mountain.

Task: mine horizontally
Description: Mine horizontally underground.
```

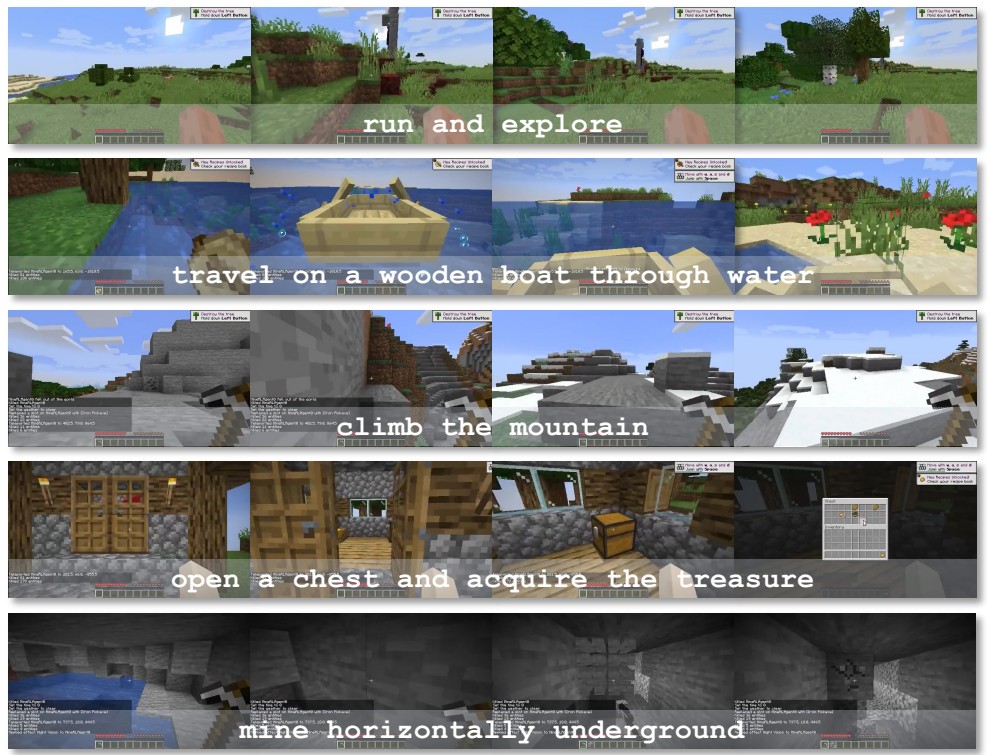

Figure 12: Examples of tasks in `explore` category.

```
Precondition: Spawn the player in a deep cave with an iron_pickaxe in the hand.
SkillAssessed: Underground navigation, tool utilization, and spatial reasoning in
    confined spaces.
Evaluation: Run away. < Break the stone. < Dig down. < Mine horizontally.

Task: travel by boat
Description: Travel on a wooden boat through water.
Precondition: Spawn the player near the sea with a wooden boat in the hand.
SkillAssessed: Aquatic travel, tool placement, and boat maneuverability.
Evaluation: Did not place the boat. < Place the boat on the water. < Board the
    boat. < Row in the water.

Task: explore the treasure
Description: Rush into a villager's home and open a chest and acquire the treasure.

Precondition: Spawn the player in front of a villager's house.
SkillAssessed: Interaction with structures, curiosity-driven exploration, and
    object acquisition.
Evaluation: Ignore the house and run away. < Open the door. < Enter the house. <
    Open the chest. < Acquire the treasure.
```

Listing 2: The environment configuration and evaluation metric for `explore` series tasks.

## H.3 CRAFT

The tasks under the `craft` category in our benchmark have been designed to shed light on an AI agent's prowess in item utilization, the intricacies of Minecraft crafting mechanics, and the nuances of various game mechanic interactions. These tasks provide a detailed examination of an agent's capability to convert materials into functional items and harness the game's various crafting and enhancement mechanics.

```
Task: craft the crafting_table
Description: Open inventory and craft a crafting table.
```

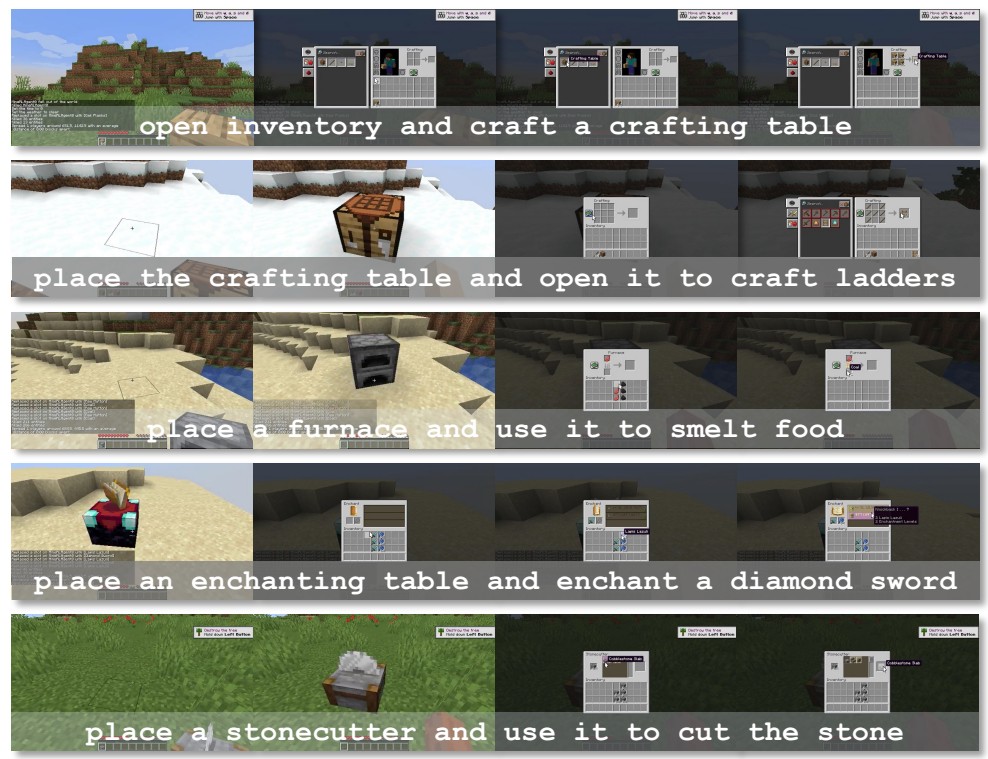

Figure 13: Examples of tasks in `craft` category.

**Precondition**: Spawn the player in the plains biome with a stack of oak_planks in the inventory.
**SkillAssessed**: Inventory management and basic crafting.
**Evaluation**: Open the inventory. < Click on the recipe button. < Click on the crafting_table. < Drag the crafting_table into the inventory.

**Task**: craft ladders
**Description**: Place the crafting table and open it to craft ladders.
**Precondition**: Spawn the player in the plains biome with a crafting_table in its main hand and a stack of oak_planks in the inventory.
**SkillAssessed**: Advanced crafting using crafting stations and recipe navigation.
**Evaluation**: Place the crafting_table on the surface. < Open the crafting_tabe. < Click on the recipe book. < Click on the ladder. < Drag the ladder into the inventory.

**Task**: enchant sword
**Description**: Place an enchanting table and use it to enchant a diamond sword.
**Precondition**: Spawn the player in the plains biome with an enchanting table in its main hand, 3 diamond swords, and 3 stacks of lapis_lazuli in the inventory.
**SkillAssessed**: Tool enhancement using enchantment stations and decision-making in choosing enchantments.
**Evaluation**: Place the enchanting_table on the surface. < Open the enchanting_table. < Place the lapis_lazuli or diamond sword. < Place the lapis_lazuli and diamond sword. < Choose any enchantment.

**Task**: smelt food
**Description**: Place a furnace and use it to smelt food.
**Precondition**: Spawn the player in the plains biome with a furnace table in its main hand, 3 stacks of mutton, and 3 stacks of coal in the inventory.
**SkillAssessed**: Food processing using a smelting furnace, raw material to product conversion, and patience in awaiting outcomes.
**Evaluation**: Place the furnace on the surface. < Open the furnace. < Place raw meat or coal. < Place both raw meat and coal. < Wait for the raw meat to be cooked. < Take out cooked meat.

**Task**: cut stone

```
Description: Place a stonecutter and use it to cut stones.
Precondition: Spawn the player in the plains biome with a stonecutter in its main
    hand, 6 stacks of stones in the inventory.
SkillAssessed: Tool enhancement using enchantment stations and decision-making in
    choosing enchantments.
Evaluation: Place the stonecutter on the surface. < Open the stonecutter. < Place
    the stones. < Select a target type of stone. < Drag stones to the inventory.
```
Listing 3: The environment configuration and evaluation metric for `craft` series tasks.

## H.4    TOOL

The tasks within the `Tool` category of our benchmark are designed to deeply investigate an AI agent's capabilities in tool utilization, precision in tool handling, and contextual application of various tools to carry out specific tasks. This category provides insights into the agent's skill in wielding, using, and exploiting tools optimally within different Minecraft scenarios.

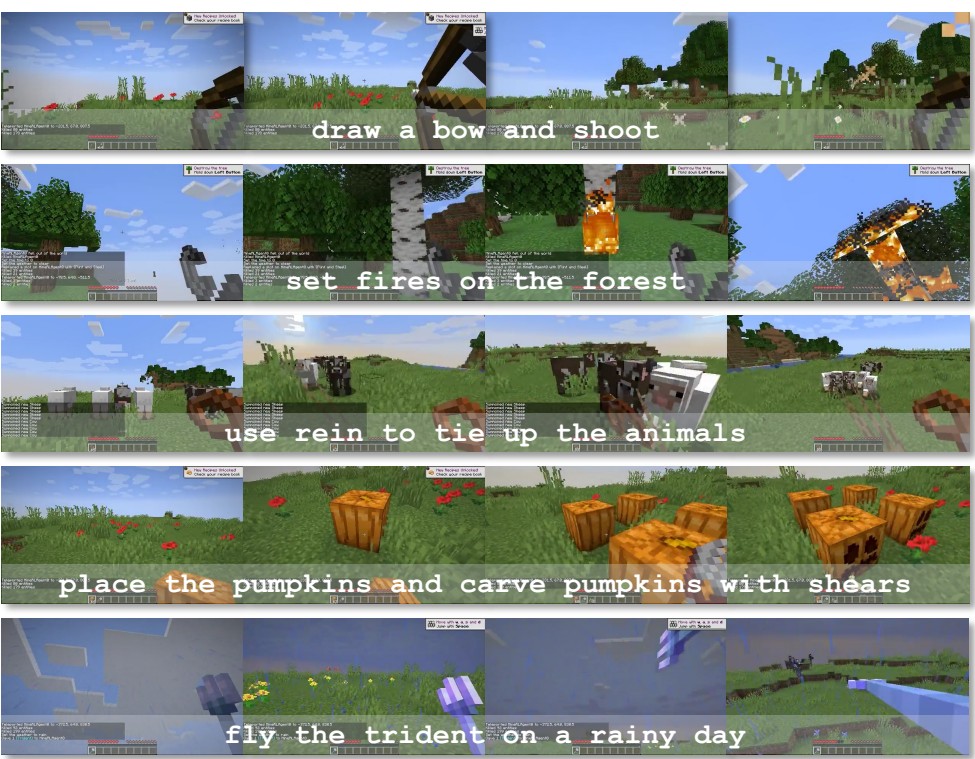

Figure 14: Examples of tasks in `tool` category.

```
Task: use bow
Description: Draw a bow and shoot.
Precondition: Spawn the player in the plains biome with a bow in the mainhand and
    a stack of arrows in the inventory.
SkillAssessed: Precision, tool handling, and projectile mastery.
Evaluation: Just run. < Draw the bow and shoot the arrow. < Hold the bow steady
    and charge up the shot before releasing the arrow.

Task: set fires
Description: Set fires on the trees.
Precondition: Spawn the player in the forest biome with a flint_and_steel in its
    main hand.
SkillAssessed: Environment manipulation and controlled chaos creation.
Evaluation: Attack the tree. < Start a fire with the flint_and_steel. < Go wild
    with the fire.

Task: lead animals
```

```
Description: Use rein to tie up the animals.
Precondition: Spawn the player in the plains biome with a stack of leads in its
    main hand. Spawn 5 sheep and 5 cows near the player's position.
SkillAssessed: Entity interaction, tool application on moving entities, and
    livestock
Evaluation: Ignore the animals and run away. < Use the rein to tie up animals.

Task: carve pumpkins
Description: Place the pumpkins and carve pumpkins with shears.
Precondition: Spawn the player in the plains biome with a shear in its main hand
    and a stack of pumpkins in the inventory.
SkillAssessed: Block placement, block modification, and crafting interaction.
Evaluation: Just run. < Place the pumpkin on the surface. < Use the shear to carve
    it. < Get a carved pumpkin.

Task: use trident
Description: Fly the trident on a rainy day.
Precondition: Spawn the player in the plains biome with a trident in the main hand,
    which is enchanted with riptide. The weather is rain.
SkillAssessed: Weather-adaptive tool utilization, motion dynamics, and advanced
    weapon handling.
Evaluation: Just run. < Use the trident to break the block. < Use the trident for
    quick movement. < Charge to throw the trident farther.
```

Listing 4: The environment configuration and evaluation metric for `tool` series tasks.

## H.5 SURVIVE

The tasks embedded within the `survive` category of our benchmark aim to analyze an AI agent's ability to ensure its own survival, adeptness in combat scenarios, and its capability to interact with the environment in order to meet basic needs. Survival, being a core aspect of Minecraft gameplay, necessitates an intricate balance of offensive, defensive, and sustenance-related actions. This category is structured to ensure a thorough evaluation of these skills.

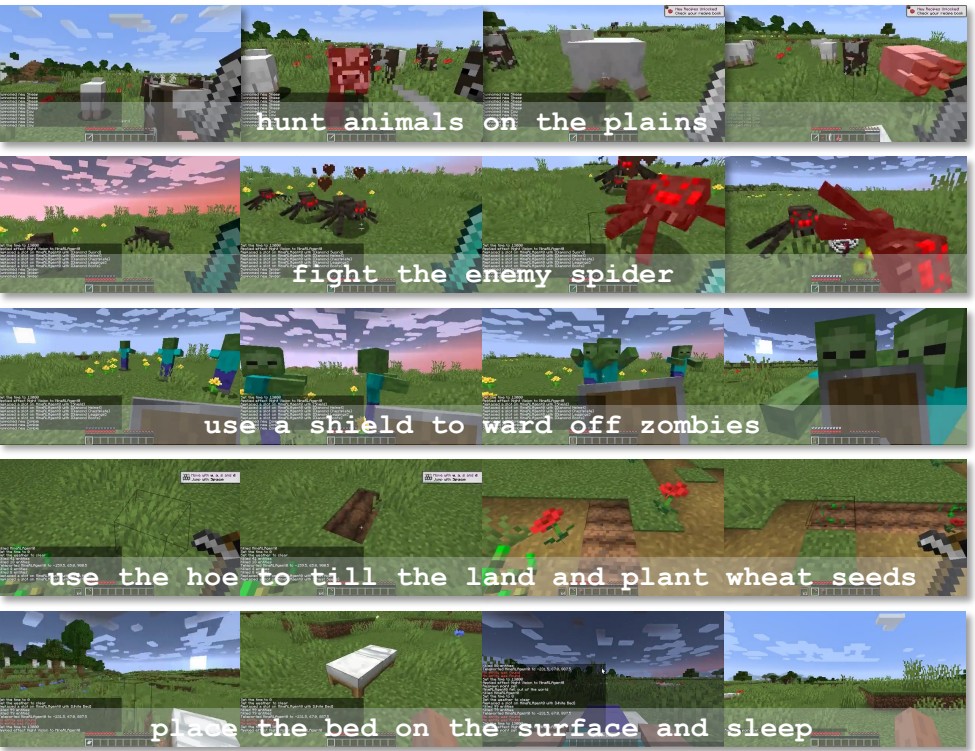

Figure 15: Examples of tasks in `survive` category.

```
Task: hunt animals
Description: Hunt animals on the plains.
Precondition: Spawn the player in the plains biome with an iron sword in the main
    hand. Spawn 5 sheep and 5 cows near the player's position.
SkillAssessed: Predator instincts, combat efficiency, and sustenance acquisition.
Evaluation: Ignore animals and run away. < Hurt animals. < Kill animals.

Task: combat enemies
Description: Fight the enemy spider.
Precondition: Spawn the player in the plains biome with a diamond sword in its
    main hand and a suite of diamond equipment. Spawn 3 spiders in front of the
    player.
SkillAssessed: Self-defense, offensive combat strategy, and equipment utilization.
Evaluation: Ignore spiders and run away. < Hurt spiders. < Kill spiders.

Task: use shield
Description: Use a shield to ward off zombies.
Precondition: Spawn the player in the plains biome with a shield in its main hand
    and a suite of diamond equipment. Spawn 3 zombies in front of the player.
SkillAssessed: Defensive tactics, tool application in combat, and strategic
    protection.
Evaluation: Ignore zombies and run away. < Use the shield to protect itself.

Task: plant wheats
Description: Use an iron_hoe to till the land and then plant wheat seeds.
Precondition: Spawn the player in the plains biome with an iron hoe in its main
    hand, and a stack of wheat seeds in the off hand.
SkillAssessed: Land cultivation, planting proficiency, and sustainable resource
    creation.
Evaluation: Just run away. < Till the land. < Plant the wheats.

Task: sleep on the bed
Description: Place the bed on the surface and sleep.
Precondition: Spawn the player in the plains biome with a white bed in its main
    hand.
SkillAssessed: Self-preservation, understanding of day-night cycle implications,
    and use of utilities for rest.
Evaluation: Just run away. < Place the bed on the surface. < Sleep on the bed.
```

Listing 5: The environment configuration and evaluation metric for `survive` series tasks.

## H.6   BUILD

The tasks within the `build` category of our benchmark are devised to evaluate an AI agent's aptitude in structural reasoning, spatial organization, and its capability to interact with and manipulate the environment to create specific structures or outcomes. Building is an integral component of Minecraft gameplay, requiring an intricate interplay of planning, creativity, and understanding of block properties.

```
Task: build pillar
Description: Build a pillar with dirt.
Precondition: Spawn the player in the plains biome with a stack of dirt in the
    main hand.
SkillAssessed: Vertical construction and basic structure formation.
Evaluation: Just run away. < Look down. < Jump and place the dirt. < Pile the dirt
    into a few pillars. < Make a really high pillar.

Task: dig three down and fill one up
Description: Dig three dirt blocks and fill the hole above.
Precondition: Spawn the player in the plains biome.
SkillAssessed: Ground manipulation and depth perception.
Evaluation: Just run away. < Look down. < Dig down three dirt blocks. < Raise the
    head. < Raise the head and use dirt to fill the hole.

Task: build gate
Description: Build an archway gate.
Precondition: Spawn the player in the plains biome with a stack of oak_planks in
    the main hand.
SkillAssessed: Symmetry, planning, and aesthetic construction.
```

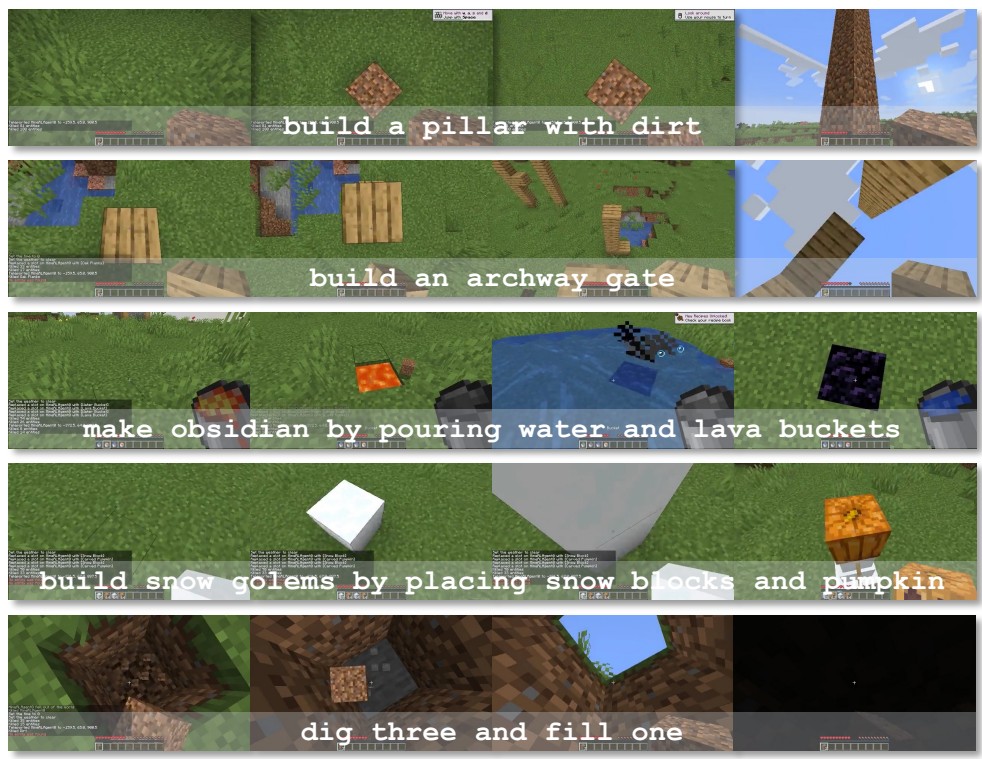

Figure 16: Examples of tasks in `build` category.

```
Evaluation: Place no plank. < Build 1 pillar. < Build 2 pillars. < Build an
    archway gate.

Task: build obsidian
Description: Make obsidian by pouring a water bucket and a lava bucket.
Precondition: Spawn the player in the plains biome with two water buckets and two
    lava buckets in the Hotbar.
SkillAssessed: Material transformation, understanding of in-game chemistry, and
    precise pouring.
Evaluation: Just run away. < Pour water or lava. < Pour both liquids. < Pour into
    a mold to make obsidian.

Task: build snow golems
Description: Build snow golems by placing two snow blocks and one carved pumpkin.
Precondition: Spawn the player in the plains biome with two stacks of snow blocks
    and two stacks of carved pumpkins in the Hotbar.
SkillAssessed: Entity creation, sequential block placement, and combination of
    multiple materials.
Evaluation: Place no block. < Place at least one kind of block. < Place both kinds
    of blocks. < Build a snow golem.
```

Listing 6: The environment configuration and evaluation metric for `build` series tasks.

# I TEXT CONDITIONING

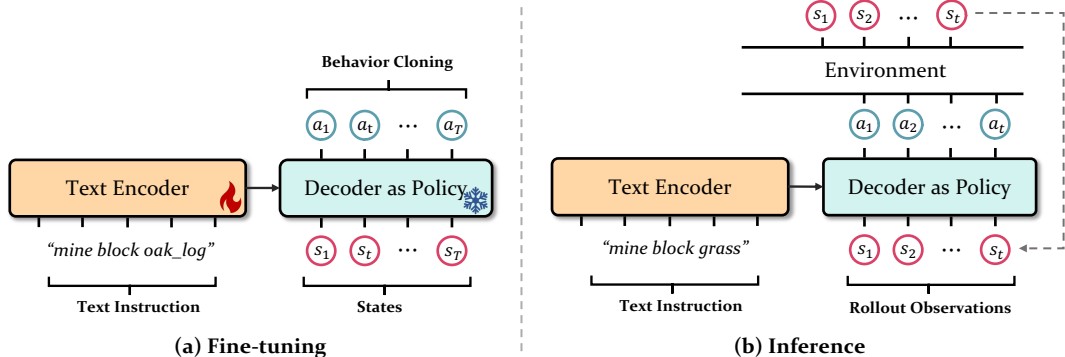

| (a) Fine-tuning | (b) Inference |

Figure 17: **Fine-tuning GROOT to understand text instructions.** We replace the original video encoder with a text encoder to embed text instructions. The text encoder is fine-tuned to align with the learned goal space. **Left:** During the fine-tuning, we freeze the learned decoder to provide the supervisory signal to train the text encoder through behavior cloning. **Right:** During the inference, we can feed forward the text instructions to drive the policy to interact with the environment.

Although video instruction has strong expressiveness, it still requires preparing at least one gameplay video for a new task. For most common tasks, such as collecting wood or stones, using natural language to specify a goal is a more natural approach. In this section, we explore the possibility of aligning the pre-trained goal space with other modal instructions, such as text instructions.

Aligning text instructions with visual instructions in goal space has been validated as feasible by Lifshitz et al. (2023). They train a conditional variational autoencoder to project text into video space after collecting 10,000 diversified text-video pairs, similar to what unCLIP did. However, the success of this alignment method depends on the pre-alignment of visual and text spaces through large-scale contrastive pre-training (Fan et al., 2022). During the training process of GROOT, we did not leverage the MineCLIP visual encoder to encode videos, instead trained goal space from scratch. On the one hand, this is because MineCLIP can only handle short videos (only 16 frames); on the other hand, it is to free our goal space from the expressiveness bounded by pre-trained VLM.

According to the above discussion, we choose to replace the video encoder in the GROOT architecture with a text encoder, BERT, and directly optimize it through behavior cloning, as shown in Figure 17. In order to keep the original goal space, we freeze the decoder and regard it as a gradient generator that extracts high-level behavioral semantics from the demonstrations. We utilize the meta information in the contractor data to generate text-demonstration pairs. For example, in the task of "collect wood", we identify the moment $t$ when event "mine_block:oak_log" is triggered in the video, and we capture the frames within the range of $[t - 127, t]$ to form a video clip, with "mine block oak log" assigned as its text, thus constructing a sample. Having been fine-tuned on these data, our model demonstrated some steerabilities in the text instruction space, as shown in Table 6. We find that the agent fine-tuned on the text-demonstration dataset shows a basic understanding of text instructions. Our method exhibits progress in tasks such as "mine grass", "mine wood", "mine stone", "mine seagrass", "pickup beef" and "mine dirt". However, it falls short in successfully completing tasks such as "mine seagrass". We speculate that this may be related to the distribution of the data, as there is much less data available for "mine seagrass" compared to the other tasks (about 300 trajectories).

***We emphasize that this experiment is very preliminary. In this experiment, the steerability of the agent fine-tuned on text instructions is still weak and it is hard to solve practical tasks.*** Given the limited diversity of text instructions in the provided contractor data, we don't anticipate the model to possess any significant level of generalization with regard to language instructions. To further verify this point, one needs to collect more diverse and higher-quality text-demonstration pairs data. Anyway, this experimental result still indicates the possibility of optimizing the upstream instruction generator by leveraging the pre-trained decoder. This creates possibilities for developing more interesting applications on GROOT. Additional discussions on text-conditioning are beyond the scope of this paper, and we will leave them for future work.

Table 6: **Text conditioning results on resource collection tasks.** Each episode lasts 30 seconds (600 frames). Statistics are measured over 10 episodes. The term "baseline" refers to the model before being fine-tuned, while "fine-tuned" refers to the final model after fine-tuning.

| Variant | mine grass ↑ | mine wood ↑ | mine stone ↑ | mine seagrass ↓ | pickup beef ↑ | mine dirt ↑ |
|---|---|---|---|---|---|---|
| baseline | 3.9 | 0.4 | 1.8 | 1.3 | 0.0 | 0.0 |
| fine-tuned | 17.3 (4.4×) | 3.7 (9.3×) | 11.5 (6.4×) | 1.2 (92%) | 0.1 | 1.3 |

## J    POTENTIAL APPLICATIONS AND INTEGRATION WITH PLANNER

GROOT is specialized in short-horizon instruction-following tasks with its goal being a video clip while LLM has demonstrated the ability to plan for long-horizon tasks in an open-world environment. For example, DEPS Wang et al. (2023b) utilizes a text-conditioned policy from Cai et al. (2023) to accomplish tasks such as mining diamonds from scratch. By integrating GROOT into the DEPS framework, it can act as a controller and assist with long-sequence tasks. However, while LLM can output language as the current goal, GROOT requires a specified video clip as its goal. Therefore, when combining GROOT with DEPS, it is necessary to use the visual language model CLIP to select the most suitable video clip based on the language goal produced by LLM.

The proposed approach involves preparing a pre-existing library of video clips $V = \{v_i\}$ that contains various actions performed by the agent in Minecraft (e.g., "chopping trees" or "mine iron ore"). When given a long-horizon instruction by LLM's Planner, it is decomposed into a series of short-horizon language tasks $\{g_i\}$. During task execution, the CLIP model is utilized to calculate the similarity between each short-horizon clip $v_i$ in the library $V$ and task $g_i$, selecting the most similar video clip as GROOT's interaction goal with the environment. Additionally, accessing a video library of Minecraft content is effortless due to the abundance of available video data on the internet.

While GROOT mainly relies on videos for input goals, LLM uses both input and output language modalities. These modalities can be aligned using a visual language model, allowing us to combine GROOT as a short-horizon control policy with an LLM-based Planner to complete long-sequence tasks.

