# OpenReview forum: "GROOT: Learning to Follow Instructions by Watching Gameplay Videos"
_ICLR.cc/2024/Conference — ICLR 2024 spotlight_

### Official Review · Reviewer_WJvG · 2023-10-25

**Soundness:** 3 good
**Presentation:** 3 good
**Contribution:** 4 excellent
**Rating:** 8
**Confidence:** 5

**Summary:**

The paper focuses on creating agents that complete tasks specified by a demonstration clip: agent is placed in the world and is given a demonstration clip of the goal, and agents' goal is to complete the task specified in the video. The paper proposes a new method for training such policy on unsupervised data (only videos, no skill/task labels needed). The results show that the proposed method outperforms the baselines by a significant margin.

**Strengths:**

Overall, considering the inclusion of a promising method and a new benchmark in a challenging environment, I recommend this paper for acceptance in ICLR. I have some hesitations around the results and comparisons (see weaknesses), which is why I steer away from strong accept, but I believe these results and contributions to be useful for ICLR community as a whole.

### Originality

The proposed idea is new, and a surprisingly simple to implement. The work done for evaluation is also novel in this exact setup. The idea of using demonstration-conditioned policies is not novel per-se, but is under-explored compared to using reward signals or simple human feedback.

### Quality

The results are solid: the paper presents a new (large) benchmark for testing models in the studied setting, and compare the proposed method against baselines, outperforming them with a clear margin. The experiments also push the method's limits with longer horizon tasks and ablate results, and also study the learned embeddings/model.

Compared to some other Minecraft results in recent years, all results were done in the "hardest" Minecraft version (one used by OpenAI VPT), without using additional helper information.

### Clarity

Paper is well structured and readable, however the amount of content makes it slow to digest (see weaknesses).

### Significance

I believe many readers in this field (conditioned agents in complex environments) would find these results useful, and would build upon the proposed method. The included benchmark is useful for future exploration to this field as well.

**Weaknesses:**

### Flawed comparisons to baselines

While STEVE-1 text/visual comparison is valid, being also goal conditioned policy, VPT models are not conditioned in any way. While it provides a decent "unconditioned baseline" result, it is not surprising that conditioning the policy improves the results. However, this is somewhat minor weakness, as there are no other baselines in the field.

### (Minor) Some uncertainty around the results

Figures 5 and 6 have rather high error bars which could indicate that results are not significantly different (see questions). Also, there are some questions around the evaluation and inference setup (see questions), which reduces my confidence in the results. However, overall, I believe the results hold and find the proposed architecture beneficial.

### (Minor) Generalization

Method was only tested on one (but complex) environment, and the tasks provide initial tools already (and the tasks are rather short-horizon). However these are more of ideas for future work than real weaknesses of the work.

### (Minor) Clarity

The paper is slightly hard to digest, but I feel this is mostly due to the amount of content (new model, human evaluation and new benchmark). Some details were kept in the appendix or missing (see questions).

I'd suggest proof-reading the paper for typos (e.g., Figure 5, "Comparision" -> "Comparison", or Page 9, "Minecraft is attracting increasing researchers". This should probably be "Minecraft is attracting _an_ increasing _number_ of researchers").

**Questions:**

1) Can you provide more details on  the human participants? Mainly following parts: how were they recruited (e.g., asked inside the lab, advertised on an online community). Did they have previous experience with Minecraft? Did they get to practice Minecraft? What instructions were they given for data collection / evaluation, during and prior the data collection? Were human participants made aware how the data collected will be used (i.e., for AI research)?
2) What are the error bars in Figure 3a? I understood the ELO rating is deterministic (given the set of comparisons/results), and does not model the uncertainty.
3) Figure 5 and 6, what are the error bars (and can you report N in the caption, I believe it is 25)? While there is trend that one is higher than the other, the scale of noise makes me wary of the results.
4) Was the night vision effect also added when VPT and STEVE agent played? They were not trained with this effect enabled, which might have skewed their results (but I do not expect huge change).
5) Do you have ablation results for the logit-substraction trick described in Appendix C.2? You highlight it as an important change, but it is hard to tell how important it is given there are no results reported for this change.
6) How did you determine 1500 pairwise to be sufficient for the ELO rankings? I assume the ranking did not change towards the end of the collection. However, I'd recommend you to also see if other metrics like TrueSkill would paint the similar picture. TrueSkill is much like ELO, but also tracks uncertainty of the rankings. See this for an open-source implementation: https://trueskill.org/


## Update 20th Nov

I have read authors' rebuttal and acknowledge they have satisfied my questions. I have kept my original review score the same (8).

---

> ### Author Response · Authors · 2023-11-19
> **Response to Reviwer WJvG - Part 1**
>
> Dear reviewer, thank you for taking the time to carefully review our paper. Thank you very much for your appreciation of the originality, quality, structure, and importance of our work, such as "The proposed idea is new, and a surprisingly simple to implement". We are also grateful for your recommendation to the ICLR community for our paper's acceptance and your belief in the potential impact of our work, as indicated by your comment "... would build upon the proposed method". We are deeply encouraged! Below, we will address the comments.
>
> ### **(Question 1) Details on the human participants.**
>
> Thank you for your question. **We have updated these details in Appendix C.3 of the paper.**
>
> >  We recruited 15 students with varying degrees of Minecraft game experience, ranging from a few hours to several years, from the Minecraft project group to conduct the evaluation. They are all familiar with the basic operations of Minecraft. Each employee was asked to label 100 matches for ELO Rating or TrueSkill Rating, for a total of 1500 matches. For each employee who is required to collect or assess gameplay videos, we ask them to first read the description of each task in the Minecraft SkillForge Benchmark completely, as well as the evaluation criteria for task completion quality. Taking the task of building a snow golem as an example, the evaluation criteria are as follows: build a snow golem $\succ$ place both kinds of blocks $\succ$ place at least one kind of block $\succ$ place no block. This enables employees to quantify video quality and ensures that all employees evaluate task completion consistently. All these employees were explicitly informed that the collected data would be used for AI research.
>
> ### **(Question 2) Explaination of error bar in Figure 3a.**
>
> Thank you for pointing that. We found that although the ranking remained relatively stable after 1500 tournaments, there was still some variation in the ELO score around the 1500-tournament mark. Therefore, we took the ELO scores corresponding to tournament numbers 1400, 1450, and 1500, respectively, and calculated their means and standard deviations to more accurately reflect the level of the agent.
>
> ### **(Question 3) Explanation of error bar in Figure 5 and 6.**
>
> Thank you for raising this concern. We have updated the number of experiments to $N=10$ in the caption.
>
> The larger variance is related to the environmental characteristics of Minecraft. Taking the task of "collect wood" as an example, we summoned the agent in a forest with complex terrain, including pools, caves, etc. We observed that when the agent accidentally fell into a pool, it took a lot of time to climb out of the water. Since the video instruction for chopping wood did not include a description of how to guide the agent to climb out of the pool, in most cases, the agent remained stuck in the pool and was thus unable to obtain any wood. If the agent did not fall into the pool, it could usually chop down a tree and obtain 4-5 blocks of wood at once, resulting in a larger variance. This phenomenon is also reflected in the STEVE-1 paper, such as its Figures 3, 4, 5, 7.
>
> ### **(Question 4) Environment configuration.**
>
> Thank you for raising this question. The environment settings during the inference, including whether to enable the night vision effect, are consistent across all baselines. Enabling the night vision effect is mainly to allow evaluators to clearly observe the agent's behavior and environmental changes, thus making accurate judgments. Additionally, players generally use torches to improve visibility in dark environments, meaning that most segments of the training data that involve mining in underground areas have relatively high levels of lighting. This is similar to the effect of the night vision effect. Therefore, we believe that this will not cause inconsistency between the training and inference processes.

---

> > ### Author Response · Authors · 2023-11-19
> > **Response to Reviwer WJvG - Part 2**
> >
> > ### **(Question 5) Ablation results for logit-substraction trick.**
> >
> > Thank you for your question. **We added the relevant experimental results in Appendix D.2** in light of your comment. We conducted experiments on four challenging tasks and found that $\lambda=1.5$ leads to 40%, 50%, 70%, and 20% absolute improvements in performance compared to $\lambda=0.0$ for "build obsidian", "build snow golem", "enchant sword", and "dig three down and fill one up", respectively. We attribute this to the fact that these tasks mostly require performing multiple consecutive actions within a small range. Therefore, stripping off the behavior of large movements using the logit-subtraction trick leads to a significant increase in the success rate. In the future, it will be meaningful to directly remove biased behaviors from the training process.
> >
> > ### **(Question 6) TrueSkill Evaluation.**
> > > How did you determine 1500 pairwise to be sufficient for the ELO rankings? I assume the ranking did not change towards the end of the collection.
> >
> > Yes, you are right. After 1500 tournaments, we found that the rankings between the various baselines no longer changed, and the ELO scores tended to stabilize.
> >
> > > However, I'd recommend you to also see if other metrics like TrueSkill would paint the similar picture.
> >
> > **Thanks for your recommendation, we have added the TrueSkill evaluation results in Appendix G.2.** We found that the ranking order of the baseline methods is consistent with that obtained using ELO rating: HUMAN $\succ$ GROOT $\succ$ STEVE-1(visual) $\succ$ STEVE-1(text) $\succ$ VPT(bc) $\succ$ VPT(fd) $\succ$ VPT(rl).
> >
> > | Baseline         | HUMAN          | GROOT        | STEVE-1(visual) | STEVE-1(text) | VPT(bc)      | VPT(fd)      | VPT(rl)      |
> > | ---------------- | -------------- | ------------ | --------------- | ------------- | ------------ | ------------ | ------------ |
> > | $\mu \pm \sigma$ | $34.2 \pm 1.0$ | $29.1\pm0.9$ | $25.8\pm0.8$    | $24.6\pm0.8$  | $22.2\pm0.8$ | $20.7\pm0.8$ | $19.2\pm0.9$ |
> >
> > ---
> > We would like to express our gratitude again for your efforts in carefully evaluating our work. Your valuable comments and positive feedback were much appreciated by us. Please feel free to share any additional comments, questions or concerns you may have.

---

> > > ### Comment · Reviewer_WJvG · 2023-11-20
> > >
> > > Thank you for your extensive answer and additional experiments! I am satisfied with these answers. I especially like the clear signal from the TrueSkill rankings; almost all models are significantly better then the ones right to them. I am still worried with the "logit-substraction trick", as it seems to contribute such performance gain while not being part of core contributions, but I agree it is a future work. Experiments in this work focused on comparing the different baselines and the proposed method.
> > >
> > > I am keeping my score as is (8), as the next step (10) is too strong for this paper. While I believe it is a good contribution to ICLR community, it does not quite reach "highlight" level.

---

> > > > ### Author Response · Authors · 2023-11-20
> > > >
> > > > Thank you for your feedback and for maintaining your score! We appreciate your thorough review and constructive comments. We will keep in mind your suggestions for future work and improvements.

---

### Official Review · Reviewer_9vuS · 2023-10-29

**Soundness:** 3 good
**Presentation:** 3 good
**Contribution:** 3 good
**Rating:** 8
**Confidence:** 3

**Summary:**

This work proposes a new agent architecture GROOT that learns goals from videos. The encoder-decoder transformer-based agent, learns from supervised videos with actions and is able to replicate the goals in reference videos. The agent GROOT is evaluated on a fairly comprehensive benchmark Minecraft SkillForge which covers a wide range of activities. Results are competitive and open up for more follow-up works in this domain.

**Strengths:**

The newly designed benchmark is a nice addition to help the evaluation of the proposed agent. It covers a wide range of different activities in the Minecraft environment, including some long-horizon tasks, building tools. Release of the evaluation benchmark is able to help the community.

The design of the model architecture intuitively makes sense.

**Weaknesses:**

There are not enough training details disclosed in the paper. Ablation on the KL loss is nice. More ablation studies on for example the number of learnable tokens would be appreciated. These experiments will further validate the robustness of the model for the task.

The training of the model still requires action input. This means that for raw video, GROOT relies on inverse dynamics model to generate pseudo action labels. The idea of an agent learning from video might have oversold the novelty of the architecture/algorithm. The attempt to sell the algorithm as unsupervised learning is repeated in the conclusion section.

The position of the figures are out of order with the experiment/result section, making it hard to navigate.

**Questions:**

Figure 3 bottom rows have both the baselines not achieving any of the task, with the first plot in row 2 having the weaker baseline outperforming the VPT fine-tuned in the experiment. Is there a particular explanation towards this?

---

> ### Author Response · Authors · 2023-11-19
> **Response to Reviewer 9vuS**
>
> Dear Reviewer, we sincerely appreciate your valuable time and recognition of our work in architectural design and benchmarking. We will release the entire benchmark in the future. Your comment "Results are competitive and open up for more follow-up works in this domain" has greatly encouraged us. Thank you for your questions, and please find our responses below, hoping to address any concern you may have.
>
> ### **(Weakness 1) Ablation on number of condition slots (learnable tokens).**
> Thank you for your question. **We have added an ablation emperiment on numer of condition slots in Appendix D.3.** We compared the performance of the model on 6 programmatic tasks with $N=1$ and $N=3$ condition slots and computed quantitative metrics for each task. As shown in the Table, we find that increasing the number of condition slots leads to a decrease in the model's performance on most tasks, except for the "explore run" task. We speculate that having more condition slots may result in a higher number of dimensions in the goal space, which makes the goal space less abstract and poorly generalizable. Therefore, we suggest that when applying GROOT to other environments, the hyperparameters should be carefully chosen based on the characteristics of the environment or using other parameter selection methods.
> | Task Name (Metric)                 | N = 1 | N = 3 |
> | ---                                | ---   | ---   |
> | explore run (distance)             | 54.0  | 59.0  |
> | build pillar (height of pillar)    | 37.6  | 13.3  |
> | collect grass (num of grass)       | 23.8  |  5.6  |
> | collect seagrass (num of seagrass) |  3.3  |  0.9  |
> | collect dirt (num of dirt)         |  6.2  |  5.4  |
> | mine stone (num of stones)         | 12.2  | 11.0  |
>
>
> ### **(Weakness 2) Emphasize the importance of inverse dynamic model.**
>
> Thanks for raising this question. The Inverse Dynamic Model is indeed critical in our paper because it provides necessary behavior labels for video data to compute the behavior cloning loss in subsequent steps. **To avoid misunderstandings, we added a section to introduce IDM in the Appendix C of the paper.** In addition, we have also emphasized this in the appropriate section of the paper, such as conclusion.
>
> Fortunately, OpenAI made an important and surprising discovery in their VPT paper that we may only need to collect a small amount of supervised data with behavior labels to train a highly accurate inverse dynamic model, even in complex environments like Minecraft. In parallel, a work presented at ECCV2022 [1] trained an accurate IDM model in the field of autonomous driving, further validating the feasibility of training IDM models in complex domains. This allows us to migrate this algorithm to other complex open environments at a lower cost.
>
> [1] Learning to drive by watching youtube videos: Action-conditioned contrastive policy pretraining.
>
> ### **(Weakness 3) Rearrange the position of the images.**
>
> Thank you for pointing that. In light of your recommendation, we have rearranged the position of the images to make them close to their descriptions.
>
> ### **(Question 1) Analysis of baselines' performance in Figure 3 (right).**
>
> Thank you for your question. As shown in the last row of Figure 3 (right), all baselines, including VPT(bc) and STEVE-1(visual), achieved low success rates on the three tasks of "enchant sword", "dig three down and fill one up", "build snow golems". This is due to the nature of these tasks. For example,"dig three down and fill one up" is a skill that only novice players would use once before passing the night. "build snow golems" is an operation that most Minecraft players are not even aware of. Therefore, the distribution of these tasks in the collected data is extremely sparse. VPT directly models the distribution of the entire dataset, making it difficult to emerge corresponding behaviors. As for STEVE-1(visual), it also performed poorly on these tasks. We speculate that these tasks are process-oriented and a 16-frame outcome video cannot fully express them. As shown in the first plot on row 2, VPT(bc) outperformed STEVE-1(visual). We speculate that this may be because VPT(bc) was fine-tuned on an early game dataset, which may contain many clips of players using furnaces to cook food. Therefore, when a furnace was placed in the agent's hand, the agent had a higher probability of exhibiting the behavior of cooking food. However, this behavior is also difficult to express with a 16-frame outcome video.
>
> ---
> We would like to extend our gratitude for your valuable feedback and queries that have helped us improve the paper. We sincerely hope that our responses have addressed your questions. We are open to answering any further queries during the discussion period, so please feel free to let us know if you have any remaining concerns or need further clarification. If we have successfully addressed your concerns, we kindly request you to consider increasing your score.

---

> ### Author Response · Authors · 2023-11-21
>
> Dear Reviewer 9vuS,
>
> We were wondering if you have had a chance to read our reply to your feedback. As the time window for the rebuttal is closing soon, please let us know if there are any additional questions we can answer to help raise the score!
>
> Best, Authors

---

> ### Comment · Reviewer_9vuS · 2023-11-22
> **Maintaining my evaluation.**
>
> Thank the authors for the clarifications.
>
> After reviewing the author's response and other reviews, it help me better understand the work. I understand that a lot of the details and experiments are not feasible to squeeze into the main paper.
>
> I am thus improving my score from 6 to 8.

---

> > ### Author Response · Authors · 2023-11-22
> >
> > Thank you so much for improving your score! We appreciate your thorough review and constructive comments!

---

### Official Review · Reviewer_f4yG · 2023-10-30

**Soundness:** 3 good
**Presentation:** 2 fair
**Contribution:** 2 fair
**Rating:** 5
**Confidence:** 3

**Summary:**

This paper proposes a method that utilizes reference videos as instructions to reduce the cost of text-gameplay annotations in constructing controllers that follow open-ended instructions. Drawing inspiration from the concept of Variational Autoencoders (VAE), the authors employ an encoder to map reference videos to the goal space, followed by a decoder acting as a goal-conditioned policy. The paper establishes a benchmark in the Minecraft, and demonstrating the effectiveness of video-based instructions compared to textual and visual instructions.

**Strengths:**

Significance: This paper addresses an interesting problem, that is, using gameplay videos to train instruction-following controllers, and validates the effectiveness of videos compared to visual information and text.

Originality: The main innovation of this paper lies in proposing a method for learning the representation of video instructions (i.e., goal space) and providing relevant theoretical derivations. Through ablation experiments, the paper demonstrates that imposing constraints on the goal space can improve performance. The visualization of the goal space is also intuitive.

Quality: This paper provides a theoretical basis for the main innovation and conducts a fair comparison with baseline methods.

Clarity: The description of the methods in this paper is clear, but some symbols lack descriptions, and some implementation details are insufficiently described, which makes the reading a bit confusing.

**Weaknesses:**

1. The modeling method for video instruction in this paper is innovative and can improve performance. However, since the low-level controller still needs to select task-related videos as instructions when performing specific tasks (Section 5.1), it cannot be well adapted to LLM-based planning agents. Although the video instructions used are generated from other biomes, it only proves the controller's generalization ability in specific tasks, rather than the generalization ability for video instructions.
- Can the authors validate the controller's generalization ability for diverse tasks by providing relevant video instructions for unseen tasks and skills? This would be of great significance for the integration of LLM-based planning agent capabilities.
- If GROOT is combined with a high-level planner, how would it construct video instructions?
- For the results in Figure 7, how are the two video instructions of GROOT specifically input to the controller? Is it directly combined and input, or is the video switched manually after reaching the specified depth?

2. Although the paper often emphasizes learning from gameplay videos, the inverse dynamic model (IDM) is based on VPT, and there is a lack of discussion on IDM. Therefore, I believe that the essence of this paper is to use task-related state sequences as instructions and train a goal-conditioned policy. Using state sequences as instructions/goals for training agents is not uncommon. Methods such as GATO[1], AD[2] directly input demonstrations as token sequences into transformers, while [3] uses VLM to extract video information. Even without imposing constraints on the goal space, agents can exhibit good performance. The authors should compare or discuss these methods.

3. As shown in Figure 5, if the video contains multiple goals, the performance may be affected. Would this lead to more stringent requirements for video quality during application?

4. In Figure 6, when the KL loss is ablated, the performance decreases. Can the authors provide a visualization of the goal space without KL loss to explain its contribution to the agent's generalization in the goal space? As a significant part of the main contribution, the KL loss needs to be analyzed in more detail.

5. Additionally, there are several unclear descriptions in the paper that may affect the understanding of the method.
- In Section 3, symbols like $\theta$ lack explicit definitions.
- In Equation 13, why does log $p(a_τ|s_{0:τ})$ depend solely on the environment dynamics?
- In Section 4.1, what does "a set of learnable embeddings" specifically refer to?
- How is the segment length T set? Is there only a single goal within each segment?
- In Equation 3, is the policy goal-conditioned?
- There is a lack of detailed description for the baselines, as well as an explanation for why VPT(rl) would achieve the worst results.

---

[1] Reed, Scott, et al. A generalist agent. 2022.

[2] Laskin, Michael, et al. In-context reinforcement learning with algorithm distillation. 2023.

[3] Di Palo, Norman, et al. Towards a unified agent with foundation models. 2023.

**Questions:**

Please refer to the detailed questions in the Weaknesses section.

---

> ### Author Response · Authors · 2023-11-19
> **Response to Reviewer f4yG - Part 1**
>
> Dear Reviewer, we thank you for your comments and suggestions. Thank you for acknowledging that the research problem we investigated is very interesting, and for recognizing the clarity and other strengths of our writing in the article. Below, we will address your comments.
>
> ### **Clarification on problem setting.**
> *We do not have predefined tasks for training*. We would like to clarify our problem setting. The problem to be addressed in this paper is **how to pre-train an instruction-following policy as well as a goal space using only large-scale video-action data or video data.** (Note that if the action data is unavailable, we need to use a pre-trained inverse dynamic model to label the video with actions.) It is worth emphasizing that these videos are recordings of human players freely playing in Minecraft, and we *do not have the task label*. Thus, there is no concept of a "pre-defined task," and there is no information such as reward functions available. Our aim is to **learn** such a goal space that expresses infinite tasks in Minecraft.
>
>
> ### **(Weakness 1, Part 1) Generalization ability.**
> > Can the authors validate the controller's generalization ability for diverse tasks by providing relevant video instructions for unseen tasks and skills? This would be of great significance for the integration of LLM-based planning agent capabilities.
>
> Thank you for your question. Our training method is somewhat similar to GPT. OpenAI did not need to define explicit tasks or categorize their collected data by task when pre-training GPT. Instead, they crawled a large amount of noisy text data from the internet and trained the model's ability to predict the next word autoregressively. After training, GPT has the ability to solve "unseen" tasks in zero-shot. Similarly, in the training process of GROOT, we do not need to define explicit tasks or cut each video into clips according to their semantics. Instead, we train the model's ability to self-imitate on a video segment. After training, by providing GROOT with a gameplay video that describes how to complete a task, it can imitate the process described in the video to complete the specified task in Minecraft.
>
> We can understand the tasks defined in Minecraft SkillForge as those that occur after GROOT has completed training, meaning that GROOT actually knows how to perform far more tasks than those included in SkillForge. As videos can be obtained from the internet in an unsupervised manner, we may not be able to find a behavior that GROOT has never seen, just as we would have difficulty finding a task that GPT has never encountered.
>
> Here, we highlight two tasks that are rarely seen in the dataset: "build snow golems" and "dig down three and fill one up". The former is only known to a few players, while the latter is only used occasionally by very beginner players. GROOT achieved a success rate of 60% in the "build snow golems" task and 30% in the "dig three down and fill one up" task, which demonstrates the extent GROOT is able to perform on long-tailed or even "OOD" tasks.
>
> ### **(Weakness 1, Part 2) Ideas of constructing video instructions when combined with a high-level planner.**
> Thank you for raising this question. In recent years, there have been a number of works [1, 2] aimed at high-level planning in Minecraft. These works unanimously suggest that a low-level controller should construct a skill library for use by higher-level planners. For example, DEPS [1] uses a pre-trained goal-conditioned policy as the underlying controller, as provided in [5]. However, training such a goal-conditioned policy still requires a large amount of high-quality, task-specific demonstrations. For GROOT, we can use relevant keywords to search on YouTube or ask Minecraft players to provide some tutorials for the skills that may be called by higher-level models. In practice, we can collect about 5 gameplay videos for each skill then select the best performing one that GROOT can imitate, save it in the database. It is easy to retrieve the gameplay video from the database when the planner needs to call some skills. This has greatly reduced the cost of building a skill library.
> **We have updated the relevant discussion in Appendix J for interested readers to check.**
>
> [1] Describe, explain, plan and select: Interactive planning with large language models enables open-world multi-task agents
> [2] Ghost in the Minecraft: Generally Capable Agents for Open-World Enviroments via Large Language Models with Text-based Knowledge and Memory
> [3] Open-world multi-task control through goal-aware representation learning and adaptive horizon prediction

---

> > ### Author Response · Authors · 2023-11-19
> > **Response to Reviewer f4yG - Part 2**
> >
> > ### **(Weakness 1, Part 3) Clarification of the results in Figure7.**
> > Thank you for your question. The two video instructions are completely independent of each other, and there is no concatenating operation between them. (If the experiments in Section 5.2 have caused any confusion, we apologize. The experiments in Section 5.2 are irrelevant to the experiments in Section 5.3.) We used a very simple logical discrimination program. At the beginning, we passed the video segment "dig down" as an instruction to GROOT(or STEVE-1), which would perform the behavior of digging down. The program would dynamically judge the agent's current height. (Here, obtaining height information is reasonable even human players needs to use the F3 function to check their current height when searching for diamonds.) When the player's height reached the 12 level, GROOT(or STEVE-1)'s input instruction was modified to "mine horizontally" (actually a gameplay video), which continued to guide the agent to explore and dig horizontally. Although the procedure that a high-level planner needs to execute in practical use may be quite complicated, it is not the emphasis of this paper. However, the way it interacts with the controller is roughly the same. We use such a simple logical discrimination program to demonstrate in the most simplified way how our GROOT works with the planner to solve long-sequence tasks.

---

> > > ### Author Response · Authors · 2023-11-19
> > > **Response to Reviewer f4yG - Part 3**
> > >
> > > ### **(Weakness 2, Part 1) Emphasize the importance of inverse dynamic model.**
> > > Thanks for raising this question. The Inverse Dynamic Model is indeed critical in our paper because it provides necessary behavior labels for video data to compute the behavior cloning loss in subsequent steps. **To avoid misunderstandings, we added a section to introduce inverse dynamic model in the Appendix C of the paper.** In addition, we have also emphasized this in the appropriate section of the paper, such as conclusion.
> > >
> > > Fortunately, VPT [1] made an important and surprising discovery that we may only need to collect a small amount of supervised data with behavior labels to train a highly accurate inverse dynamic model, even in complex environments like Minecraft. In parallel, a work presented at ECCV2022 [2] trained an accurate IDM model in the field of autonomous driving, further validating the feasibility of training IDM models in complex domains. This allows us to migrate this algorithm to other complex open environments at a lower cost.
> > >
> > >  [1] Baker, B., Akkaya, I., Zhokov, P., Huizinga, J., Tang, J., Ecoffet, A., ... & Clune, J. (2022). Video pretraining (vpt): Learning to act by watching unlabeled online videos. Advances in Neural Information Processing Systems, 35, 24639-24654.
> > >  [2] Learning to drive by watching youtube videos: Action-conditioned contrastive policy pretraining.
> > >
> > > ### **(Weakness 2, Part 2) Discussions of related works.**
> > > Thanks for pointing out these related works, AD[1], Gato[2], and UniAgent[3].
> > > We agree that the Transformer architecture has gradually become a common modeling method in decision control tasks. However, we believe that our method is conceptually different from GATO, AD, and UniAgent in terms of what we aim to learn/model.
> > >
> > > As for the AD algorithm, it mainly focuses on how to use the Transformer to model the training process of reinforcement learning algorithms. It can be viewed as a meta-reinforcement learning algorithm. It requires a large number of historical trajectories generated during reinforcement learning training as well as reward signals related to the tasks. However, designing reward functions for each task in Minecraft is impractical because the number of tasks is endless. In addition, during the inference process, the Transformer needs to repeat execution with the environment many times to obtain a good policy, which is infeasible in some dangerous environments. For example, in Minecraft, a single death may lead to the entire game's progress being wiped out. Therefore, the AD algorithm is not suitable for solving the problem encountered in this paper.
> > >
> > > As for the Gato model, although it has achieved good results in the multi-domain, multi-modal, and multi-task fields, its training still requires pre-defined tasks, which are learned by collecting a large amount of high-quality expert teaching data under the specific tasks. Actually, we do not have task labels for our gameplay videos. Therefore, GATO is not suitable for this environment.
> > >
> > > Regarding the UniAgent model, it needs to interact with the environment in real-time, which restricts its ability to scale up to learning large-scale skills. We believe that this model can achieve good performance in life-long learning settings; however, the focus of this paper is to learn the open-ended instruction-following ability from a large number of offline video data. Therefore, UniAgent is not suitable for solving this problem.
> > >
> > > **We have taken your comment into account and put the related discussion in the "Related Works" section.**
> > >
> > > [1] In-context reinforcement learning with algorithm distillation.
> > > [2] A generalist agent.
> > > [3] Towards a unified agent with foundation models.

---

> > > > ### Author Response · Authors · 2023-11-19
> > > > **Response to Reviewer f4yG - Part 4**
> > > >
> > > > ### **(Weakness 3) Clarification of Figure 5.**
> > > > Thanks for your question. The experiments in Figure 5 demonstrate a bonus discovery of our learned goal space: concatenating multiple goals enables GROOT to solve multiple tasks simultaneously to some extent. However, we also found that this may have the side effect of decreasing the performance of completing individual tasks. This is acceptable because GROOT was not trained on concatenated videos. Providing higher-quality data should lead to improved performance in the end. The discussion of how to further improve GROOT's performance on concatenated videos is beyond the scope of this manuscript and will be left to future work.
> > > >
> > > > ### **(Weakness 4) Visualization of learned goal space.**
> > > > Thank you for raising the question. We have updated the visualization of the goal space for GROOT with and without KL regularization in Figure 4. It can be seen that both of them demonstrate certain semantic understanding capabilities in clustering. The clustering effect is slightly better with KL regularization, but the difference is not very significant. Intuitively, KL loss is mainly used to bring the distribution of the posterior latent closer to the distribution of the prior latent, thereby enhancing the abstraction capability of the encoder and improving the generalization ability of the model. However, the generalization ability does not have a strong correlation with the clustering effect on the goal space of videos within the dataset, which is confirmed by our quantitative results, please refer to Figure 6.
> > > >
> > > >
> > > > ### **(Weakness 5) Clarity and Completeness of Descriptions.**
> > > >
> > > > - **Clarification of parameters $\theta, \phi$ in Section 3:** $\theta$ denotes the parameters for policy $\pi_\theta(a_t | s_{0:t}, g)$, prior $p_\theta(g | s_{0:t})$, and inverse dynamic model $p_\theta(a_{\tau} | s_{0:\tau+1})$. $\phi$ denotes the parameters for the posterior $q_\phi(g | s_{0:T})$.
> > > >
> > > > - **Clarification of $p(a_\tau | s_{0:\tau})$:** Thank you very much for pointing this out. This was a typo on our part. It actually describes a distribution within the dataset that is irrelevant to the goal space and goal-conditioned policy that we wish to learn.
> > > >
> > > > - **Clarification of "a set of learnable embeddings":** This is a set of special tokens designed to summarize the semantic information of the video. We took inspiration from recent transformer-based methods, such as ViT [1], which introduced an extra learnable "classification token" to summarize category information in images, and DETR [2], which used a set of learnable tokens called "object queries" to extract semantic and position information of objects in images.
> > > >
> > > >     [1] AN IMAGE IS WORTH 16X16 WORDS: TRANSFORMERS FOR IMAGE RECOGNITION AT SCALE.
> > > >     [2] End-to-End Object Detection with Transformers.
> > > >
> > > > - **Clarification of segment length "T":** T is a hyperparameter that is related to the maximum context length that TransformerXL processes at one time. In this work, we set T to 128, to be consistent with the structure setting of VPT. On the one hand, this is due to limited computational resources, as a larger T would consume a considerable amount of GPU memory. On the other hand, a context length of 128 already has strong expressive power and can represent most short-horizon tasks.
> > > >
> > > > - **Is there only a single goal within each segment?** Each video segment of length T is considered a complete behavior, even if it contains multiple steps, and is encoded as a single goal embedding.
> > > >
> > > >
> > > > - **Clarification of Equation 3:** We apologize for any confusion caused by the equation in question, as it was a typo on our part. The policy is indeed conditioned, and we have updated the correct equation in our paper. The formulation should be $\pi_\theta(a_t | s_{1:t}, {g})$, where $g \sim q_{\phi}(\cdot | s_{0:T})$ .
> > > >
> > > > - **Clarification of the performance of VPT(rl):**  The poor performance of VPT (rl) is due to the fact that it is a strategy specifically fine-tuned for the diamond-mining task, and does not accept goals as input. Therefore, its performance is poor in multi-task environments other than on the diamond-mining task chain (such as digging and crafting). Considering the limited existing baselines in the field of Minecraft, we chose steerable baselines STEVE-1 (visual) and STEVE-1 (text), as well as non-steerable baselines VPT (bc), VPT (fd), and VPT (rl). **In light of your comment, We have provided a detailed introduction to the baseline used in our paper in Appendix F.1**.
> > > >
> > > > ---
> > > >
> > > > We would like to extend our gratitude for your valuable feedback and queries that have helped us improve the paper. We sincerely hope that our responses have addressed your questions. We are open to answering any further queries during the discussion period, so please feel free to let us know if you have any remaining concerns or need further clarification. If we have successfully addressed your concerns, we kindly request you to consider increasing your score.

---

> ### Author Response · Authors · 2023-11-21
>
> Dear Reviewer f4yG,
>
> We were wondering if you have had a chance to read our reply to your feedback. As the time window for the rebuttal is closing soon, please let us know if there are any additional questions we can answer to help raise the score!
>
> Best,
> Authors

---

> > ### Comment · Reviewer_f4yG · 2023-11-22
> >
> > I appreciate the author's response, which has addressed my main concerns, but there are still some doubts.
> >
> > Although "may not be able to find a behavior that GROOT has never seen," the data can be artificially divided or produced. For example, given some unreasonable behavior composed of video examples, can GROOT maintain similar behavior? This is crucial for the applicability of the method.
> >
> > Using videos as instructions and IDM is the contribution of VPT, so I think the difference between GROOT and GATO is the addition of the KL divergence constraint in the representation space.
> >
> > Although the authors have proposed some ideas for combining with high-level planners, many special treatments are still needed in the application, such as using search engines or filtering videos, which limits the applicability.
> >
> > After the authors revised the paper, the clarity of the paper has been improved, so I will raise the score to 5.

---

> ### Author Response · Authors · 2023-11-23
>
> Dear Reviewer,
>
> We are very glad that we were able to solve your major problems and greatly appreciate that your score has improved. We hope that we can further clarify any doubts you may have.
>
> ---
>
> > Although "may not be able to find a behavior that GROOT has never seen," the data can be artificially divided or produced. For example, given some unreasonable behavior composed of video examples, can GROOT maintain similar behavior? This is crucial for the applicability of the method.
>
> We defined a task of "burning oneself to death using lava," which is a very unreasonable behavior. We found that GROOT performed well on this "unreasonable" task, with 4 out of 5 episodes being successful. This further demonstrates that our method can generalize well to tasks outside the training distribution.
> We have put the videos here: https://anonymous.4open.science/r/lava_ano-0C5B
>
> ---
>
> > Using videos as instructions and IDM is the contribution of VPT, so I think the difference between GROOT and GATO is the addition of the KL divergence constraint in the representation space.
>
>
> The video instructions mentioned in this paper refer to a way of specifying a task, which describes how a task should be completed in the form of a video.
>
> - VPT is a model obtained by behavior cloning a large-scale trajectory dataset, which can be represented as $a_t \sim \pi(a_t | s_{1:t})$. Therefore, **it does not have the ability to follow instructions and naturally cannot follow video instructions**. We have tried to append a video instruction (chop tree) prefix as a prompt to the state sequence input to VPT. However, this did not prompt VPT to obtain the corresponding behavior, and the actual result was that VPT almost completely ignored this prompt.
>
> - For GATO, as mentioned in section 5.2 of its paper, **it requires collecting a large amount of expert demonstration data for new tasks and finetuning on these data to generalize to new tasks**.
> The following is the content from section 5.2 of the GATO paper:
> > Ideally, the agent could potentially learn to adapt to a new task via conditioning on a prompt including demonstrations of desired behaviour. However, due to accelerator memory constraints and the extremely long sequence lengths of tokenized demonstrations, the maximum context length possible does not allow the agent to attend over an informative-enough context. Therefore, to adapt the agent to new tasks or behaviours, we choose to fine-tune the agent's parameters on a limited number of demonstrations of a single task, and then evaluate the fine-tuned model's performance in the environment.
>
>   In the GATO paper, achieving the same performance as the BC algorithm on a new task requires collecting 5000 demonstrations for that task. In contrast, for GROOT implemented with our encoder-decoder architecture, we only need to collect a minimum of 1 video to generalize to a new task, and we do not require any additional finetuning process. This means that we do not have to worry about "catastrophic forgetting" caused by finetuning on a new task.
>
> ---
>
> > Although the authors have proposed some ideas for combining with high-level planners, many special treatments are still needed in the application, such as using search engines or filtering videos, which limits the applicability.
>
> For humans, searching for strategy guides or instructional videos recorded by other gamers on the internet is a common practice when playing games. Therefore, we believe that using a search engine or filtering video data to build a skill database is a natural method. A powerful generalist agent should have the ability to retrieve knowledge and expand its skills on the internet (although this is beyond the scope of this research).
> In addition, we also provide other ideas for integrating GROOT with a high-level planner. For example, by collecting text-video pairs, we can align the text instruction modality with the instruction space of GROOT. We conducted a preliminary experiment in **Appendix I** as an example for your reference.
> However, we would like to clarify that the main problem that this paper addresses is how to train an agent to understand video as a form of task description. There are many tasks that are difficult to describe in language, making video a natural choice for description.
>
> ---
>
> We hope that the above explanations can resolve doubts you may have.

---

### Official Review · Reviewer_anV3 · 2023-10-31

**Soundness:** 3 good
**Presentation:** 4 excellent
**Contribution:** 3 good
**Rating:** 8
**Confidence:** 4

**Summary:**

The authors propose GROOT, a video-goal-based instruction following agent. GROOT consists of an encoder-decoder architecture, with a video encoder for goal-conditioning and a causal transformer for the action policy. This architecture enables the agent to be conditioned on videos as flexible multi-task goals, rather than the more limited spaces of language or final states as studied in prior work. GROOT shows impressive performance when tested against prior methods for training agents in Minecraft using the proposed Minecraft SkillForge benchmark.

**Strengths:**

- The overall paper is well written and the method is clearly presented. The proposed method is simple, and providing the ability to condition more flexibly on multi-step video-based goals is a meaningful contribution in the space of developing generalist agents, as there is an abundance of video-based data available online to learn from.
- The experiments seem thorough, with clear improvements compared to prior methods on a suite of different complex tasks within Minecraft. The paper considers both quantitative skill improvements as well as qualitative goal space analyses, showing that the learned goal space has likely captured some meaningful semantics.

**Weaknesses:**

- While the multi-step video representation for goals is more expressive than other prior Minecraft agents (that use language or outcome videos), having to provide a video to condition on can be difficult to do in practice if we do not already have access to similarly representative videos, and as the authors note, training the video-based goal space is challenging. On the other hand, it is much easier to describe a desired goal in language. It would be interesting to see if this method can be adapted flexibility to condition on either visual or text based goals, as in STEVE-1.
- Nit: confusing notation – changing from $p_\theta$ to $p_\psi$ in the KL term

**Questions:**

It’s interesting that conditioning on multiple videos concatenated together can drive the agent to accomplish multiple tasks simultaneously, and the embedding space seems to be qualitatively meaningful. Do the authors think there would also be a benefit to constructing synthetic clips concatenating multiple tasks and training on them as well? Even if the source video settings look very different (potentially making goal-space learning harder), it might be an interesting way of generating large amounts of interesting synthetic, multi-task data.

---

> ### Author Response · Authors · 2023-11-19
> **Response to Reviewer anV3**
>
> We would like to express our sincere appreciation to the reviewer for their positive feedback on the clarity and simplicity of our proposed method, as well as its potential contribution to the development of generalist agents. Thank you for your valuable insights and encouragement!
>
> ### **(Weakness 1) Additional experiment on text conditioning.**
> Thank you for your question. We believe the video instruction space is more general and some behaviors are more naturally expressed with video. It is the focus of our paper.
>
> Meanwhile, we also agree on the importance of conditioning policy on text instructions. Although video instruction has strong expressiveness, it still requires preparing at least one gameplay video for a new task. For some tasks, such as collecting wood or stones, using natural language to specify a goal is a more natural approach. **We have conducted the text-conditioning experiment, please see Appendix I**.
>
> Considering that it is not feasible for us to collect text-video data with as much diversity as STEVE-1 within a short period of time, and we did not directly utilize the initial alignment space provided by MineCLIP during GROOT training. Therefore, we chose a new alignment method, which is to directly replace the video encoder with a BERT architecture text encoder (randomly initialized) and freeze the decoder. We optimize the text encoder using behavior cloning.
>
> We utilize the meta information in the contractor data to generate text-demonstration pairs. For example, in the task of "collect wood", we identify the moment $t$ when event "mine\_block:oak\_log" is triggered in the video, and we capture the frames within the range of $[t-127, t]$ to form a video clip, with "mine block oak log" assigned as its text, thus constructing a sample. Having been fine-tuned on these data, our model demonstrated some steerabilities in the text instruction space, as shown in the table below. (The term "baseline" refers to the model before being fine-tuned, while "fine-tuned" refers to the final model after fine-tuning. )
>
> | Variant    | mine grass    | mine wood    | mine stone    | mine seagrass| pickup beef        | mine dirt  |
> | ---------- | ------------- | ------------ | ------------- | ------------ | ---------------- | ---------- |
> | baseline   | 3.9           | 0.4          | 1.8           | 1.3          | 0.0              | 0.0        |
> | fine-tuned | 17.3          | 3.7          | 11.5          | 1.2          | 0.1              | 1.3        |
>
> We find that the agent fine-tuned on the text-demonstration dataset shows a basic understanding of text instructions. Our method exhibits progress in tasks such as "mine grass", "mine wood", "mine stone", "mine seagrass", "pickup beef" and "mine dirt". However, it falls short in successfully completing tasks such as "mine seagrass". We speculate that this may be related to the distribution of the data, as there is much less data available for "mine seagrass" compared to the other tasks (only about 300 trajectories).
>
> We emphasize that this experiment is very preliminary. In this experiment, the steerability of the agent fine-tuned on text instructions is still weak and it is hard to solve practical tasks. Given the limited diversity of text instructions in the provided contractor data, we don’t anticipate the model to possess any significant level of generalization with regard to language instructions. To further verify this point, one needs to collect more diverse and higher-quality text-demonstration pairs data. Anyway, this experimental result still indicates the possibility of optimizing the upstream instruction generator by leveraging the pre-trained decoder. This creates possibilities for developing more interesting applications on GROOT. Additional discussions on text-conditioning are beyond the scope of this paper, and we will leave them for future work.
>
> ### **(Weakness 2) Confusing notation.**
>
> Thank you for raising this question. We apologize for the confusion caused. It was a typo on our part. Our intention was to clarify that the two video encoders do not share the same set of parameters. In reality, we should use $p_\theta(g | s_{0:t})$ instead of $p_\psi(g | s_{0:t})$. We have corrected this in the paper.

---

> > ### Author Response · Authors · 2023-11-19
> > **Response to Reviewer anV3 - Part 2**
> >
> > ### **(Question 1) Explore concatenated video as training data.**
> >
> > Thank you for proposing the very interesting idea of training GROOT directly on multi-task concatenated videos to enhance its ability to perform multiple tasks simultaneously. We believe this idea is reasonable and natural. Although the original GROOT demonstrated some ability to perform multiple tasks on concatenated videos, its performance on individual task decreased. We attribute this to inconsistencies between the distribution of training data and the distribution of test data. GROOT trained on concatenated videos would likely alleviate this issue. However, due to resource and workload constraints, we will leave this intriguing idea for future work. Thank you again for your valuable suggestion.
> >
> > ---
> > We would like to express our gratitude again for your efforts in carefully evaluating our work. Your valuable input and positive feedback were much appreciated by us. Please feel free to share any additional comments, questions or concerns you may have.

---

### Author Response · Authors · 2023-11-19
**General Response**

We thank all reviewers for their insightful comments and acknowledgment of our contributions. We highlight the major contributions of our work as follows:

- We are "addressing an interesting problem" (Reviewer f4yG), i.e. creating agents that complete tasks specified by a demonstration clip. We proposed "a new idea and is surprisingly simple to implement" (Reviewer WJvG, Reviewer anV3), which does "a meaningful contribution in the space of developing generalist agents" (Reviewer anV3). This paper "provides a theoretical basis for the main innovation" (Reviwer f4yG). "Many readers would build upon the proposed method." (Reviewer WJvG, Reviewer 9vuS).

- We evaluate GROOT "on a fairly comprehensive benchmark Minecraft SkillForge which covers a wide range of activities" (Reviewer 9vuS), "the included benchmark is useful for future exploration to this field as well" (Reviewer WJvG), "open up for more follow-up works in this domain" (Reviewer 9vuS).

- Our experiments "seem thorough, with clear improvements" (Reviewer anV3), "the results are solid", "all results were done in the "hardest" Minecraft version" (Reviewer WJvG), "Many readers in this field (conditioned agents in complex environments) would find these results useful" (Reviewer WJvG).

We’ve revised our manuscript per the reviewers’ suggestions (highlighted in red in the uploaded revision pdf). Detailed responses to each reviewer’s concerns are carefully addressed point-by-point. Below summarize the major updates we’ve made:

- Visualization of learned goal space w/o KL loss (**Figure 4**).

- Citations to related works (**Related Works**).
> [1] In-context reinforcement learning with algorithm distillation.
> [2] A generalist agent.
> [3] Towards a unified agent with foundation models.

- Emphasis the importance of inverse dynamic model in our method (**Page 3, Conclusion**).

- A section introduces the background knowledge of the inverse dynamic model in **Appendix C**.

- An ablation study on the condition scale $\lambda$ when using logit-substraction trick during the inference. (**Appendix D.2**)

- An ablation study on number of condition slots in **Appendix D.3**.

- A comprehensive introduction of baselines in **Appendix F.1**.

- Results of TrueSkill evaluation in **Appendix G.2**.

- Details of human participation in **Appendix G.3**.

- A text conditioning experiment in **Appendix I**.

- A section describes a potential way to integrate GROOT with high-level planner in **Appendix J**.


We believe our method and benchmark could make a timely contribution to the decision making community and would like to involve in further discussions if any question is raised.

Best,
Authors

---

### Meta-Review · Area_Chair_Rn9A · 2023-12-07

**Metareview:**

The authors present a goal-conditioned behavioral cloning approach to learn to perform tasks in Minecraft given a reference video. The proposed method uses a transformer architecture and is simpler than previous work in this space, specifically STEVE-1, in that it does not rely on text. However, this also means that it cannot execute textual descriptions of tasks and relies on the availability of a reference video. Moreover, the method relies on the availability of an inverse dynamics model, weakening the claim of being able to learn from unlabeled videos. The method is described clearly, the empirical evaluation is extensive, and the results are impressive. While the proposed benchmark seems comprehensive, automatic scoring available in the simulated environment would have been preferable over human ratings for reproducibility purposes. All in all, the reviewers agree that this paper is a valuable contribution to the reinforcement learning literature.

**Justification For Why Not Higher Score:**

Only minimal ablations, requires availability of inverse dynamics model, no automatic evaluating for proposed benchmark

**Justification For Why Not Lower Score:**

Simple and well-presented method with strong empirical performance

---

### Decision · Program_Chairs · 2024-01-16

Accept (spotlight)